



# Uncertainties in breakup markers along the Iberia-Newfoundland margins illustrated by new seismic data

Annabel Causer[1], Lucía Pérez-Díaz[1,2], Jürgen Adam[1] and Graeme Eagles[3]

[1]Earth Sciences Department, Royal Holloway University of London, Egham, TW20 0EX, United Kingdom
[2]Department of Earth Sciences, Oxford University, Oxford, OX1 3AN, United Kingdom
[3]Alfred Wegener Institut, Helmholtz Zentrum für Polar und Meeresforschung, Bremerhaven, Germany

*Correspondence to*: Annabel Causer (annabel.causer.2017@live.rhul.ac.uk)

**Abstract.**

Plate tectonic modellers often rely on the identification of "break-up" markers to reconstruct the early stages of continental separation. Along the Iberian-Newfoundland margin, so-called "break-up markers" include interpretations of old magnetic anomalies from the M-series, as well as the "J-anomaly". These have been used as the basis for plate tectonic reconstructions on the belief that these anomalies pinpoint the location of first oceanic lithosphere. However, uncertainties in
the location and interpretation of break-up markers, as well as the difficulty in dating them precisely, has led to plate models that differ in their depiction of the separation of Iberia and Newfoundland.

We use newly available seismic data from the Southern Newfoundland Basin (SNB) to assess the suitability of commonly used break-up markers along the Newfoundland margin for plate kinematic reconstructions. Our data shows that basement associated with the younger M-Series magnetic anomalies is comprised of exhumed mantle and magmatic
additions, and most likely represents transitional domains and not true oceanic lithosphere. Because rifting propagated northward, we argue that M-series anomaly identifications further north, although in a region not imaged by our seismic, are also unlikely to be diagnostic of true oceanic crust beneath the SNB. Similarly, our data also allows us to show that the high amplitude of the J Anomaly is associated to a zone of exhumed mantle punctuated by significant volcanic additions, and at times characterised by interbedded volcanics and sediments. Magmatic activity in the SNB at a time coinciding with M4
(128 Ma), and the presence of SDR packages onlapping onto a basement fault suggest that, at this time, plate divergence was still being accommodated by tectonic faulting.

We illustrate the differences in the relative positions of Iberia and Newfoundland across published plate reconstructions and discuss how these are a direct consequence of the uncertainties introduced into the modelling procedure by the use of extended continental margin data (dubious magnetic anomaly identifications, breakup unconformity
interpretations). We conclude that a different approach is needed for constraining plate kinematics of the Iberian plate pre M0 times.
## 1 Introduction

Over the past decade, plate tectonic modellers working on divergent settings have focused their efforts on better-constraining
the early stages of continental separation, partly driven by the oil and gas industry's move to more distal and deeper
exploration targets. As of today, bridging the gap between the onshore and offshore geological evolution of rifted continental
margins still presents a challenge, due to the difficulty in unequivocally interpreting the complex geology of extended
continental margins.

When studying divergent settings, the onset of seafloor spreading is often based on so-called "breakup markers" that
originate in tectonic interpretations made along the extended continental margins. Identified and mapped from geophysical
data, these features include depositional unconformities (e.g. Pereira et al., 2011; Soares et al., 2012; Decarlis et al., 2015),
packages of landward dipping reflectors (e.g. Keen and Voogd, 1988), and seismic amplitude changes in the top-of-basement
surface (e.g. Tucholke et al., 2007), interpreted to mark the change from continental to oceanic crust. These interpretations
are utilised as the basis for many computer-generated plate reconstructions, which are in turn highly susceptible to
uncertainties associated with the interpretation and mapping of said breakup markers.

Uncertainties of this kind, and their impact on tectonic reconstructions, have been illustrated by, for example, the alternative
scenarios proposed in the literature for the movements of the Iberian plate between the Late Jurassic to Early Cretaceous.
Rotational poles derived from interpretations of the location of the continent-ocean boundary (COB), for example, result in
overlaps of known continental crust along the Iberia-Africa plate boundary (e.g. Srivastava and Verhoef, 1992). Such
overlaps are not present in kinematic models built on the basis of magnetic anomalies, which assume Iberia moves together
with Africa for much of this time period (e.g. (Sibuet et al., 2012). A further study, constituting a combination of magnetic
seafloor anomalies and on-land palaeomagnetic data, shows the Iberia-Africa boundary to be more complex (Neres et al.,
55  2013).

The West Iberia and Newfoundland margins are considered by many as the type-example for magma-poor passive rifted
margins (Boillot et al., 1995; Whitmarsh and Wallace, 2001; Reston, 2007; Tucholke and Sibuet, 2007; Péron-Pinvidic and
Manatschal, 2009). The continental margins are the result of Late Triassic to Early Cretaceous rifting and separation of the
North American and Eurasian plates. This pair of conjugate margins has been the focus of more than 40 years of intense
research, including extensive geophysical surveying and drilling campaigns as part of the Ocean Drilling Programme (ODP)
and Deep Sea Drilling Project (DSDP) (e.g. Whitmarsh and Sawyer, 1996; Wilson et al., 1996). Research has revealed the
margins' tectonic asymmetry and the gradual proximal to distal transition from regions of highly extended continental crust





to zones of exhumed mantle at times intruded by pre or post-breakup magmatic intrusions. Despite this, the detailed plate
kinematics, the age of distinct rift episodes, the timing of final breakup, and the significance of pre-existing structures and
lithological heterogeneity are still heavily debated. The difficulty in identifying, mapping and dating the COB along this pair
of conjugate margins is evident in the wide range of candidate COBs suggested in the literature (Fig. 1) (i.e. Eagles et al.,
2015 and refs. therein). The age of final break-up and formation of first oceanic crust is particularly uncertain. Drilling
results and breakup unconformity identifications date the onset of seafloor spreading at the Aptian-Albian transition (113
Ma) (Tucholke and Sibuet, 2007; Boillot et al., 1989). This is significantly younger than the age of the oldest isochrons
interpreted from magnetic reversal anomalies (M20-145 Ma to M0-120 Ma) offshore Iberia (Srivastava et al., 2000) (Fig. 1).
The interpretation of these anomalies in terms of M-series isochrons is disputed. Although interpreted by some studies as
markers of first oceanic lithosphere (e.g. Vissers and Meijer, 2012; Sibuet, et al., 2004), others have shown that they may
instead be associated with igneous bodies located within zones of exhumed mantle (e.g. Sibuet et al., 2007; Sibuet et al.,
2012).

Here we describe and interpret a number of previously unpublished 2D seismic profiles imaging the regional tectonic
structure and crustal architecture of the Southern Newfoundland margin from the shelf to the deepwater oceanic basin. Our
interpretations underline the structural and kinematic complexity of the transitions between continental and oceanic crust at
the Iberia-Newfoundland conjugate margins that contribute to the challenges faced by plate modellers when reconstructing
this pair of conjugate margins.

Furthermore, review a number of published studies in order to examine the uncertainties of available plate kinematic
reconstructions of the Iberia-Newfoundland conjugate margin. We do this by (a) examining the locations, within our new
seismic data, of "breakup markers" commonly used by said studies and (b) utilising these published rotation schemes to
reconstruct conjugate margin transects into their pre-drift positions, examining the consequences of choosing alternative
rotation parameters.

## 2 Study area – tectonic evolution and controversies

The formation of the Iberian - Newfoundland conjugate margins are primarily a result of a series of northward propagating
Late Triassic to Early Cretaceous rifting episodes (Manatschal and Bernoulli, 1999; Alves, et al., 2009; Wilson et al., 2001).
Progressive extension, and final localization of the divergent plate boundary at a mid-ocean ridge led to the separation of the
North American and the Iberian plates. Unlike the classic textbook examples of passive margin architecture, continental and
oceanic crust are not juxtaposed along these margins, but separated by a very wide continent-ocean transition zone (150-180
km, (Eagles et al., 2015) (Fig. 1). Geophysical research into the Iberian - Newfoundland margins has, to an extent, illustrated
the gradual change from continental crust through regions of exhumed continental mantle and into purely oceanic crust (e.g.



Dean et al., 2015). Although transition zones like this have been widely studied over the past decade (*e.g.* Whitmarsh and Wallace, 2001; Manatschal et al., 2001; Pérez-Gussinyé and Reston, 2001; Péron-Pinvidic and Manatschal, 2009; Mohn et al., 2012), the identification so-called break-up features, which cannot be confidently attributed to either crustal type, renders kinematic reconstructions based on them difficult and susceptible to large uncertainties. In literature, this transition is often
referred to as continent-ocean transition zone (COTZ).

The complex architecture of the Iberian - Newfoundland margins is the result of a sequence of extensional deformation episodes beginning with an initial "wide-rift" phase during late Triassic-earliest Jurassic times (Manspeizer, 1988; (Manatschal and Bernoulli, 1998;  Tucholke et al., 2007; Péron-Pinvidic et al., 2007). This was followed by the localisation
of extension and related crustal thinning along the distal part of the future margins, which resulted in the exhumation of subcontinental mantle rocks within the transition zones, leading up to seafloor spreading sometime in the Early Cretaceous (Manatschal and Bernoulli, 1999; Dean et al., 2000; Malod and Mauffret, 1990; Péron-Pinvidic et al., 2007; Tucholke et al., 2007). The exact age of the onset of seafloor spreading is controversial. Some suggest initiation in the Barremian (Whitmarsh and Miles, 1995; Russell and Whitmarsh, 2003), and others the Valanginian (Wilson et al., 2001) or perhaps as
late as the Aptian – Albian boundary (Tucholke et al., 2007b) based on interpretation of a breakup unconformity marking the onset of seafloor spreading (Tucholke et al., 2007b; Péron-Pinvidic et al., 2007; Mauffret and Montadert, 1987; Boillot et al., 1989).

One of the difficulties in reconstructing the separation of the Iberian - Newfoundland margins is presented by the complex
kinematic history of the Iberian plate. Although currently part of the Eurasian plate, the Iberian plate moved independently between the Late Jurassic and sometime in the Paleogene (Fig. 2). During the Late Jurassic to Early Cretaceous, the Iberian plate was separated from the African, North American and European plates by divergent plate boundaries (Le Pichon and Sibuet, 1971) (Fig. 2, a-c). During Aptian time, relative motions between the African, Iberian and Eurasian plates underwent a period of re-organisation (Roest and Srivastava, 1991; Pinheiro et al., 1996;  Rosenbaum et al., 2002;  Seton et al., 2012;
Tavani et al., 2018). It is broadly accepted that the Iberian plate undertook an anticlockwise rotation of around 35° with respect to the Eurasian plate, resulting in the opening of the Bay of Biscay along its northern margin (Fig. 2, b-c). Considerable controversy still exists as to the exact nature, timing and consequences of this rotation, with conflicting scenarios having been proposed by authors based on interpretations of geological and geophysical observations (Olivet et al., 1984; Srivastava et al., 2000; Gong et al., 2008; Vissers and Meijer, 2011). Kinematic reconstructions can be split into two
end member groups. In one, the Bay of Biscay is depicted as having opened in a scissor-like fashion, with the hinge of the scissors located in south-eastern corner of the Bay of Biscay (Srivastava et al., 2000) (as shown in Fig. 2d). In the other, opening happens in a left lateral manner (*Olivet, 1996*). The anticlockwise rotation of Iberia as recorded in paleomagnetic data (*e.g.* Gong et al., 2008) is most closely replicated by models depicting a scissor-type opening (Srivastava et al., 2000). However, models like these imply significant compression further east along the IB-EUR plate boundary (*e.g.* Masson and



Miles, 1984; Matthews and Williams, 1968; Roest and Srivastava, 1991; Schoeffler, 1965; Sibuet and Collette, 1991; Sibuet, and Srivastava, 1994; Srivastava et al., 1990, 2000), which is not supported by field geology (Lagabrielle et al., 2010; Tugend et al., 2014). The presence of numerous bodies of sub-continental mantle rocks exposed along the North Pyrenean Zone (*Bodinier et al., 1988;* Lagabrielle et al., 2010*;* Vauchez et al., 2013) instead suggest the formation of extensional basins during the Cretaceous. Some authors have interpreted these basins as having formed in a back-arc setting resulting

from the subduction of older oceanic lithosphere from north of Iberia beneath Europe (Sibuet et al., 2004; Vissers and Meijer, 2012). Alternatively, the opening of the Bay of Biscay can be interpreted as the result of strike-slip motion between Iberia and Europe, along the North Pyrenean Fault (*e.g. Olivet et al., 1996*). Although in this model the fit of Iberia and Eurasia, derived by fitting the prominent regional magnetic J Anomaly, deteriorates to the north, it is favoured by many (Stampfli et al., 2002; Jammes et al., 2009; Handy et al., 2010).


Partial closure of the Bay of Biscay between Late Cretaceous and Oligocene times led to the formation of the Pyrenees (Bullard et al., 1965; Van der Voo, 1969; *Muñoz, 2002;* Sibuet et al., 2004; McClay et al., 2004; Gong et al., 2008) (Fig. 2, e-f). In the early Miocene, the plate boundary between Iberia and Eurasia became inactive and the Iberian plate was incorporated into the Eurasian plate (Van der Voo and Boessenkool, 1973; Grimaud et al., 1982; Sibuet et al., 2004; Roest

and Srivastava, 1991; Vissers and Meijer, 2012) so that the boundary between Eurasia and Africa ran south of Iberia and into the North Atlantic along the Azores-Gibraltar Fracture Zone (AGFZ) (Le Pichon and Sibuet, 1971; Sclater et al., 1977; Grimaud et al., 1982; Olivet et al., 1984*;* Roest and Srivastava, 1991; Zitellini et al., 2009)). The present-day AGFZ (Fig. 1) is a complex plate boundary that accommodates relatively small differences between Eurasian-North American and African-North American seafloor spreading rates and directions along the Mid-Atlantic Ridge in the forms of minor extension at its

western end (Searle, 1980), right-lateral strike-slip along its middle reach, and transpression in the east (*e.g.* Srivastava et al., 1990; Grimison and Chen, 1986; Jiménez-Munt and Negredo, 2003).

## 2.1 Break-up markers along the Iberian – Newfoundland margins

It is generally agreed that statistical fitting of fracture zone trends and oceanic isochrons determined from magnetic

anomalies is the most accurate method of modelling the relative motions of plates for the last 200 Ma. This is a consequence of the relatively small locational error and relatively high interpretational confidence compared to other geological and geophysical markers (Müller et al., 2008; Seton et al., 2012; Pérez-Diaz and Eagles, 2014). Despite this, the presence of magnetic reversal anomalies is not of itself diagnostic of crustal type, particularly along passive margins with wide transitional zones, such as the Iberian – Newfoundland margins. Within COTZs, it is possible that magnetic anomalies

resulting from the presence of intrusive igneous bodies within the upper crust or exhumed sub-continental mantle can be erroneously attributed to basaltic oceanic crust (*e.g.* Cannat et al., 2008). Similarly, oceanic crust formed at mid-ocean ridges

that are overlain by a significant thickness of sediment (Levi and Riddihough, 1986) or formed at ultra-slow spreading centres (Roest and Srivastava, 1991; Jokat and Schmidt-Aursch, 2007) may not give rise to strong magnetic signatures.

Accordingly, whilst some researchers have interpreted magnetic anomalies as isochrons dating back to Late Jurassic (Chron M20, 146 Ma) to model relative motions of the Iberian and North American plates (Srivastava et al., 2000), their utility can be disputed by contradictory geological evidence from drill core data. At Site 1070 on the Iberian margin (Fig. 1), for instance, serpentinised peridotite was drilled from the location of a magnetic anomaly that had been previously defined in terms of seafloor spreading at the time of chron M1 (~125 Ma; Whitmarsh et al., 1996; Tucholke and Sibuet, 2007).

Numerous seismic surveys off both the Iberian and Newfoundland margins interpret the presence of transitional crust oceanwards of M0 (120 Ma), the youngest of the M-Series isochrons (Shillington et al., 2006; Dean et al., 2015b; Davy et al., 2016).

Several other M-Series isochrons have been interpreted along the North Atlantic margins from magnetic anomalies that are

often characterised by a somewhat subdued (<100 nT amplitude; Fig. 3b) magnetic signature. Although their sources too are debated, and sometimes suggested to lie within domains of exhumed mantle and thinned continental crust (Russell and Whitmarsh, 2003; (Sibuet, J et al., 2004) their apparent symmetry across the rift and parallel trend with respect to the continental margins has led many researchers to interpret them as indicators of the presence of old oceanic lithosphere. The uncertainties in the origin and interpretation of these anomalies also contribute to the generally large set of discrepancies

between plate kinematic reconstructions of Iberia, and in understanding the development of the Bay of Biscay in Late Jurassic to Early Cretaceous times (*e.g.* Srivastava et al., 1990; Whitmarsh and Miles, 1995; Srivastava et al., 2000; Barnett-Moore et al., 2016). For example, tectonic models using the M0 anomaly (125 Ma) result in a gap between eastern Iberia and Europe, the closure of which is difficult to reconcile with geological and geophysical data from the Pyrenees (Van der Voo, 1969; Gong et al., 2008; Lagabrielle et al., 2010; Tugend et al., 2014).


### 2.1.1 The "J" Anomaly

In addition to the interpretations of M-Series isochrons, a number of researchers have used a further regional magnetic lineation, known as the J anomaly, as a kinematic marker of the onset of seafloor spreading.

First acknowledged by Pitman and Talwani, (1972), the J anomaly is a high-amplitude anomaly identifiable on each side of the Southern North Atlantic Ocean south of the Galicia Bank and Flemish Cap regions (Fig. 3a). Based on its high amplitude and apparent symmetry across the rift, many have favoured the use of the J Anomaly over the M-Series as a kinematic marker. As a result, the J Anomaly has formed a basis for many plate kinematic reconstructions of the Iberia-Newfoundland conjugates (e.g. Srivastava et al., 1990, 2000; Sibuet, et al., 2004).




The amplitude, from trough to peak, of the J Anomaly is generally 500 – 600 nT in the South Newfoundland Basin (SNB) and conjugate Tagus Abyssal Plain (TAP) *(Tucholke et al., 1989),* reaching maxima of around 1000 nT over the southeast Newfoundland Ridge and conjugate Madeira Tore Rise (Fig 3b-c). The J Anomaly coincides with a structural step in the basement in the TAP (Tucholke and Ludwig, 1982) and with discontinuous basement ridges in the SNB (*Tucholke et al.,*
*1989).*

The origin and subsequent significance of the J anomaly has been interpreted in two ways in published literature. The first of these interpretations suggests that the J anomaly is the oldest magnetic isochron of true oceanic origin formed by seafloor spreading and representative of the beginning of the M-series magnetic anomalies (Keen et al., 1977; Sullivan, 1983;
Klitgord and Schouten, 1986). It may be interpreted as a superposition anomaly formed by spreading during the periods of isochrons M0 - M1 (*Rabinowitz et al., 1978; Tucholke and Ludwig, 1982*) or M0 – M4 (Whitmarsh and Miles, 1995), (Fig. 3b-c). In both cases, the J anomaly is seen as the boundary between first formed oceanic crust and exhumed mantle (Reston and Morgan, 2004).

The alternative interpretation of the J anomaly (Bronner et al., 2011), suggests that it expresses magmatic basement ridges dating from the Late Aptian (120 – 113Ma) during the time immediately preceding steady-state seafloor spreading. Both the unusually high amplitude and variable width of the J anomaly are explained by Bronner et al., (2011) as being the result of the interplay between excess surface magmatism and the locations of underplated bodies at depth. The apparent northward decrease in J anomaly amplitude and distance to chron C34 are interpreted as evidence for a northward propagating breakup.
Agreeing with this line of interpretation, Nirrengarten et al., (2017) go on to question its validity as an indicator of first seafloor spreading processes.

## 3 Dataset and Methods

A high-resolution plate kinematic model generated using seafloor spreading data (unequivocal oceanic magnetic anomalies
and fracture zone traces) would provide the ideal framework within which to investigate the evolution of the Iberia-Newfoundland passive margins. A well-constrained rotation scheme could be used to rotate regional seismic transects across both conjugate margin segments back into their paleopositions at the time of breakup to generate a virtual rift-spanning seismic transect at the time of continental break-up. This, in turn, would make it possible to investigate further how the processes related to continental breakup are recorded in the sedimentary architecture of the conjugate Iberia-Newfoundland
margins, as well as the suitability of some suggested breakup markers such as the M-Series or J anomaly as the basis for



kinematic models. However, in the North Atlantic such a kinematic model does not yet exist independently of previous interpretations of presumed-conjugate pairs of seismic profiles.

Available two-plate models built using seafloor spreading data allows us to robustly reconstruct the paleopositions of Iberia and Newfoundland only as far back to the first known isochron of undisputed oceanic origin (C34, 84 Ma) (Fig. 4). However, the incompletely-known extent of so-called ¨transitional¨ crust along the extended continental margins of the southern North Atlantic means that it is not possible to identify conjugate seismic transects on the basis of this two-plate reconstruction. Reconstructing older time slices and the break-up position on the basis of less-controversial seafloor spreading data is possible, but requires a more complex four-plate model in which the motions of Iberia and Newfoundland

are modelled in conjunction with those of the African and Eurasian plates (Causer et al. *in prep*).

Here, we describe and interpret a number of previously unpublished regional 2D seismic profiles in the SNB. The discussed seismic data were obtained from TGS-NOPEC's Southeast Grand Bank data set, which comprises some 34 2D seismic lines covering a combined area of 55,995 km$^2$. The lines discussed were acquired in 2014 using a 31.25 m shot point interval,

providing high resolution images of the crustal structure offshore Newfoundland. They extend from the continental slope, through highly-extended continental crust and into exhumed mantle domains. None of these seismic lines extend far enough oceanward to image acoustic basement that can be confidently attributed to true oceanic crust. They do, however, image transitional crust previously associated with the J anomaly (M4 – M1, Whitmarsh and Miles, 1995*).*

Unfortunately, the conjugate TGS Iberian margin 2D seismic dataset (Fig. 4) offshore Portugal does not extend far enough through the COTZ and into the distal domain to directly image crust associated with the younger M-Series (M10 – M0) isochron interpretations, where breakup markers have been interpreted along the conjugate margin. For this reason we have also re-examined a previously-published seismic profile (IAM5) *(see* Pinheiro et al., 1992; Afilhado et al., 2008; Neves et al., 2009).


The stratigraphic framework of the SNB has not been investigated in detail as part of this study. Due to the lack of drilling data, sediments have been grouped into Synrift 1, Synrift 2, Breakup-sequence, and Post-Rift packages based on seismic-stratigraphic observations. Synrift 1 corresponds to a sedimentary sequence that formed during fault-controlled extension, and is characterised by reflectors which mimic changes in basement structure, often short in length and at times chaotic, and

onlapping structural highs. Synrift 2 is instead characterised by more continuous reflections, arising from what we interpret as infill strata deposited between the end of fault-controlled rifting and onset of seafloor spreading, also known as "sag sequence" (Masini et al., 2014). Based on its high amplitude and continuous nature, we consider our Breakup sequence to mark the rupture of the lithosphere and onset of seafloor spreading, which we later tentatively date as taking place near the Aptian – Albian boundary (*e.g.* Mauffret and Montadert, 1987;  Boillot et al., 1988; Pinheiro et al., 1992; Tucholke et al.,



2007; Péron-Pinvidic et al., 2007), Although new research (Alves, and Cunha, 2018) in the conjugate Tagus Abyssal Plain (TAP) proposes the presence of two break-up sequences, the first of which initiated in Berriasian times, (145 Ma) our new seismic dataset does not allow us to repeat such an interpretation. Finally, post-rift strata are found overlying a prominent unconformity. They have been dated at DSDP Site, 398, on the Iberian margin (Fig. 1), as Cenomanian in age (Wilson et al., 1989; Alves, et al., 2003; Soares et al., 2012).


## 4 Results

### 4.1 Line A – Southern South Newfoundland Basin

Line A, located in the southern South Newfoundland basin, is a 444 km long margin-scale 2D seismic section, which images the entire crust beneath the Grand Banks area and offshore Newfoundland. Part of this line is shown in figure 5. This 2D
seismic section extends from the continental slope, through the COTZ into the distal domain.

The crust of the continental shelf beneath the Southern Grand banks is tectonically thinned by a crustal scale rift margin fault seen in the landward part of the profile between 2 and 6-7 s TWT (Fig. 5). Its hanging wall is deformed by numerous landward-dipping intra-rift faults with variable offsets. At depth this large fault is traceable to around 10 s TWT, coinciding
with our interpretation of the seismic Moho.

More distally, the margin is characterised by a series of domino-style rotated fault blocks, bounded by landward dipping faults of varying displacements (Fig. 5b). At depth, these faults seem to terminate against a high amplitude reflector traceable to depth. This high amplitude reflector can be traced to the top basement and interpreted as an exhumation fault
marking the distal extent of thinned continental lithosphere. Oceanward of this point, the basement is deformed by a series of alternating landward and oceanward dipping normal faults. This change in seismic character of the basement and its coincidence with the high amplitude reflector can be interpreted as the transition from highly extended continental crust to exhumed mantle. Landward of this location, the continental crust in the rift basin has been thinned progressively via landward dipping intra-rift faults and larger oceanward dipping faults, possibly detached at depth (Fig. 5a-b). However,
eastward of the high-amplitude reflector the imaging of acoustic basement is poor due to the presence of high-impedance post-rift strata.

In the most seaward part of the profile, high amplitude reflectors are traceable within what we interpret as a volcanic edifice (Fig 5c). Within it, reflectors dip in opposing directions, which may be a result of velocity pull-up (e.g. Magee et al., 2013).
Short discontinuous reflectors within the volcanic edifice are observed to on-lap on to syn-rift 1 strata and the interpreted top of the exhumed mantle. Although sediments associated with break-up and post-rift sequences also on-lap this syn-rift 1 /

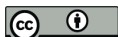

basement high, their seismic character is noticeably different. On-lapping reflectors within the volcanic edifice are shorter, brighter than and not as planar as those observed in the breakup and post-rift sequences. Accordingly, we interpret the internal high-amplitude reflectors as sills (Fig. 5c). We have also tentatively identified a potential hydrothermal vent dyke, marked by distorted seismic imaging underneath mounded seismic highs *(e.g.* Planke et al., 2005). Imaging beneath the edifice is poor, rendering interpretations of the underlying basement difficult.

## 4.2 Line B – Central South Newfoundland Basin

Line B, located in the central South Newfoundland Basin images a 264 km long crustal transect from unequivocal continental crust beneath the landward continental shelf, through highly extended continental crust in the COTZ, and into a zone of exhumed mantle with magmatic additions (Fig. 6).

The proximal part of the margin is characterised by numerous parallel oceanward-dipping normal faults following a staircase-like pattern. Their vertical extents are difficult to map with certainty. Some of these faults are seen to terminate downwards against a high amplitude reflector, which we interpret as a deep-seated landward dipping detachment fault originating at the basinward limit of continental lithosphere (Fig. 6a). Oceanward of this high amplitude reflector, the transition from highly extended continental crust to zones of exhumed mantle is marked by a smoother seismic characteristic of top basement.

In the exhumed mantle zone, a prominent basement high bisects the breakup sequence. The internal structure of the high is poorly imaged, making interpretations within it challenging (Fig. 6b). Landward of this high, a series of large basement faults bound a relatively-symmetrical 80 km wide sub-basin infilled with a thick syn-rift sedimentary sequence. Towards the seaweard part of the profile we interpret a package of seaward dipping reflectors (Fig. 6c), the top of which is marked by a high amplitude reflector. This package coincides with the interpreted location of the J Anomaly. Here, by analogy to drilled margins with similar characteristics (e.g. the south Australian margin, Ball et al., 2013), we suggest the acoustic basement to comprise a mixture of sediments and lava flows. Laterally, SDRs are seen to onlap onto a fault, perhaps indicating a degree of control by extension processes on magmatism (Fig. 6c).

## 4.3 Line C – Northern South Newfoundland Basin

Line C, located in the northernmost part of the South Newfoundland Basin is a 444 km long section which images the continental margin across the Grand Banks and offshore Newfoundland. Figure 7 shows a 180 km long oceanward segment of this seismic line, focusing on the continental shelf, highly extended continental crust and the COTZ.



At the base of the continental slope, which is characterised by a series of oceanward dipping faults, a landward dipping high-
amplitude reflector can be traced to a depth equivalent of 10 s TWT. Oceanward, the basement is characterised by regularly
spaced landward-dipping domino-style rotated fault blocks (Fig. 7a), above which we identify the presence of sedimentary
packages corresponding to syn-rift 1, syn-rift 2 and the breakup sequence.

As before, we tentatively interpret the transition between extended continental crust and transitional crust from the
smoothing of top basement. The COTZ is presumed to be floored by exhumed mantle, as recovered at sites 1276-1277
(Tucholke and Sibuet, 2007) further north in the Northern Newfoundland Basin (NNB). Within our interpreted region of
exhumed mantle, individual fault blocks are no longer interpretable. The prominent basement high shown in figure 7b may
be interpreted as a serpentinite diapir, as seen elsewhere within the Iberian Abyssal Plain and offshore the Galicia Bank
region (*e.g.*(Boillot et al., 1980, 1995)


### 4.4 IAM5 – Tagus Abyssal Plain

The wide-angle 350 km long seismic profile IAM5 images crust from the continental slope into the distal domain of the
Tagus Abyssal Plain (TAP) (Fig.9). Although previously described in detail in the literature, *(e.g.* Pinheiro et al., 1992;
Afilhado et al., 2008; Neves et al., 2009), we take this section into consideration in order to provide an Iberian conjugate to
the new seismic profiles described previously.

IAM5 is characterised by large oceanward-dipping and smaller landward-dipping basement faults in the COTZ, some of
which propagate upwards into 'undifferentiated' syn and post-rift sequences. A rise in basement toward the ocean is
observed some 160 km from the base of the continental slope. Here, fault blocks still consistently dip toward the continent.
Additionally in this distal domain, a high amplitude reflector is traceable above top basement, to 6s TWT. Although the syn
and post-rift breakup sequences are undifferentiated, the presence of sediments older than Base Cenozoic has not been
interpreted within this high (*see* Neves et al., 2009).

### 5 Discussion

The Iberia-Newfoundland margins have been extensively surveyed and studied over the past decade. The three seismic lines
presented here, across the previously poorly-documented Southern Newfoundland Basin (SNB), further illustrate the
complexity of this conjugate margin and are interpretable within the context of the existing and growing literature on
extended continental margin processes. We interpret these lines as extending from the continental shelf, through highly
extended continental crust and into distal deepwater basin characterised by the presence of exhumed mantle.

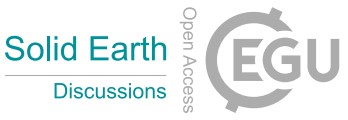


Our interpretations of the geological and structural history of the SNB also allow us to speculate about the origin of magnetic anomalies previously interpreted as diagnostic of oceanic lithosphere and extensively used as grounds upon which to base plate tectonic reconstructions of the North Atlantic.

**5.1 Magnetic isochron interpretation: M-series and J-Anomaly**

Some authors (e.g. Srivastava and Tapscott, 1986; Srivastava et al., 1990, 2000) identify the presence of M-Series magnetic reversal isochrons from magnetic anomalies recorded along the Newfoundland margin, attributing them to the presence of oceanic lithosphere. Our results do not support such an interpretation. Instead, along both lines A (Fig. 5) and B (Fig. 6) these anomalies (M1-M4) are sourced within zones of exhumed mantle which, in places, may be intruded by magmatic

additions of uncertain age. In Line B (Fig. 6), the interpreted M-Series isochrons coincide with the high-amplitude oceanward dipping reflectors that we interpret as SDR packages of interbedded volcanics and sediments. The formation of these features is usually associated with mantle dynamics during plate rupture rather than the formation of steady-state igneous crust (*e.g.* Keir et al., 2009; Yamasaki and Gernigon, 2009). Here, they may indicate the "onset" of magmatic-driven extension (Tugend et al., 2018) preceding the establishment of seafloor spreading and production of true oceanic lithosphere.

The volcanic edifice, sills and feeder dykes in Line A (Fig. 5) may also be coeval with the final stages of plate rupture.

Our interpretations align with those of Russell and Whitmarsh, (2003) and Sibuet et al., (2004) who attribute the subdued amplitudes of the Newfoundland margins' magnetic anomalies as indicative of source bodies in highly-extended continental crust and exhumed mantle, rather than the upper layers of a 'standard' 7 km-thick oceanic crust.


Our seismic Line B (Fig. 6) images crust associated with the J Anomaly in the SNB. The anomaly coincides with an area of interpreted interbedded sedimentary and igneous packages, which are on-lapping a basement fault. This might indicate that, at the time of magmatism, plate divergence was still controlled by tectonic faulting and the transition to seafloor spreading had not yet occurred. Although we acknowledge that the limited quantity of new data available to us is not, on its own,

sufficient to draw a complete picture, it suggests that the J anomaly does not represent a boundary between purely oceanic lithosphere and exhumed mantle transitional domains (*e.g.* Reston and Morgan, 2004), but instead that its source lies within or on the latter.

Although our results suggest that M-series magnetic anomaly isochrons within the Newfoundland margin do not originate

from purely oceanic lithosphere, they can be used to estimate the minimum possible age of the basement underlying them. Based on this, we suggest that the Newfoundland margin may have been magmatically-influenced since the Early Aptian (coinciding with M4, ~128 Ma) (Fig. 5), earlier than previously thought.





According to Bronner et al., (2011) the J Anomaly results from Late Aptian (120 – 103 Ma) magmatism, preceding seafloor

spreading. They suggested that northward propagating magmatism from which the J Anomaly originates began in the Northern Central Atlantic and was restrained at the Newfoundland Fracture Zone for 10 Myrs before reaching the NNB in the Iberian-Newfoundland rift at the Aptian-Albian transition (112 Ma). Our results suggest a slightly different timing, with magmatic activity present in the SNB at a time coinciding with M4 (128 Ma), some 6-8 Ma younger than that proposed by Bronner et al., (2011).


Further north (*e.g.* Tucholke et al., 2007; Bronner et al., 2011; Nirrengarten et al., 2017)*,* ODP drilling of rocks associated with the J Anomaly in the NNB revealed a similar assemblage of exhumed mantle and intrusive and extrusive mafic rocks. The drilling results suggested that magmatic activity had been persistent from ~128 Ma (M4) to ~70 Ma (Jagoutz et al., 2007).


Although the J anomaly may be associated with events immediately preceding first seafloor spreading, these events are neither instantaneous in time nor isochronous along the margin, which renders the J Anomaly unsuitable as a kinematic marker.

**5.2 Conjugate pair matching**

The wide range of processes interpretable from our new data and previous studies of the Iberia-Newfoundland margins illustrates a degree of asymmetry that makes it impossible to unequivocally identify conjugate pairs of seismic transects from their geometric and stratigraphic characteristics alone. An alternative approach could be to select conjugates by rotating margin-wide seismic lines into coincidence at pre-drift times. However, the results of doing this are strongly dependent on

the choice of rotation scheme and their inherent uncertainties. Figure 8 illustrates the wide range of pre-rift positions resulting from seven published plate kinematic models for Barremian times (Sibuet and Collette, 1991; Rowley and Lottes, 1988; Labalis et al., 2010; Seton et al., 2012; Greiner and Neugebauer, 2013; Srivastava et al., 1990). Plate reconstructions to younger time slices are unsuitable for identifying conjugates because of the significant underlap they result in between the seismic surveys either side of the ocean. Similarly, full-fit reconstructions back to early Jurassic times result in large overlaps

of the extended continental margins (Fig. 8).

Seton et al's. (2012) reconstruction (Fig. 8, b2) is based on an 'extreme-oceanic' interpretation, with magnetic isochron picks in the sequence back to M20 (Srivastava and Tapscott, 1986; Srivastava et al., 2000). This model keeps Iberia fixed to Africa throughout Barremian times. Alternatively, the model of Greiner and Neugebauer, (2013) (Fig. 8, b1), relies on the magnetic

dataset of Srivastava et al., (2000) alone to produce best-fitting reconstructions of M-Series isochrons interpreted from dense
magnetic data off Newfoundland and sparser data off Iberia. In contrast, prior to chron M0, Srivastava et al's., (1990) (Fig. 8, b3) relies more strongly on seismic interpretations of conjugate changes in basement characteristics, conjugate fracture zones, and conjugate COB segments.

The reconstruction of Seton et al., (2012) results in significantly more overlap of the COTZ envelopes than that of Greiner and Neugebauer, (2013). Overlaps in the COTZ suggest that the extended continental margins had not yet reached their present-day widths at this time. The early stages of continental separation, as described by these models, are subject to significant uncertainty, resulting from (a) the assumption that M-series anomalies are of oceanic origin and (b) the difficulty in interpreting subdued magnetic signals. This is illustrated by the differences in the reconstructions produced by the models,

shown in figure 8, b1 and b2. Despite the differences between the models of Greiner and Neugebauer, (2013) and Seton et al., (2012), both suggest Line C as a conjugate to IAM5 prior to seafloor spreading (Fig. 9).

Alternatively, the model by Srivastava et al., (1990) suggests a conjugate pair consisting of lines B and IAM5 (Fig 9). Their rotation scheme is derived from a model in which structural markers are used to constrain the position of Iberia during the

Barremian, most notably Keen and Voogd's (1988) COB, which they interpreted to coincide with a prominent landward dipping reflector (the L reflector, see Reid, 1994). The use of this feature shifts Iberia's palaeo-position 50 – 100 km further south than that modelled using identified magnetic isochrons alone.

The validity of the 'L' reflector as a breakup marker can, however, be questioned on the basis of the huge variety of

alternative COB interpretations published before and since Keen and Voogd's, (1988) study, which in this region differ by up to 200 km (Eagles et al., 2015). More specifically, Funck et al., (2003) identified the L Reflector offshore Flemish Cap to lie well inboard of the COTZ within the continental slope. We tentatively interpret a high landward dipping reflector traceable into the continental shelf in our Line C (Fig. 7), similar to the described 'L' Reflector thought to mark the COB.

Discriminating between "good" and 'bad' reconstructions on the basis of the transects they reunite is clearly challenging. In the case discussed here, no strong arguments can be made regarding which of our new seismic lines (Line B or Line C) is the more likely conjugate to IAM5 based on their structural and stratigraphic characteristics. Neither line displays features which can be solely attributed to an upper/lower plate setting in asymmetric margins *(e.g.* Lister et al., 1986). . The proximal domains of both Line B and C in the SNB are characterised by progressive continental lithosphere thinning by tectonic

faulting, in places observed to terminate against large continent-dipping detachment faults. Faulting of continental lithosphere can also be observed on the Iberian side in line IAM5, although in this case detachment surfaces are not imaged. Across the interpreted transitional domains, exhumed mantle, diapirs and extrusive flows are present in Lines B and C but absent in line IAM5, where underplating has been suggested instead, although its age is uncertain (Mauffret et al., 1989; Peirce and Barton, 1991; Bronner et al., 2011; Pinheiro et al., 2004). The Madeira Tore Rise, located at the distal end of





IAM5, results from alkaline magmatism post-dating breakup, which may have also resulted in the formation of volcanic
edifices such as that seen in Line A in the SNB.

These observations illustrate the challenge of discriminating between "good" and "bad" rotation schemes on the basis of the
conjugate transects they produce. This challenge could be greatly eased if informed by robust plate models built from high-
confidence data with quantified uncertainties.

## 5 Conclusions

In this paper we have presented and described three new seismic transects from the Southern Newfoundland Basin, and used
them to discuss the validity of widely used so-called breakup markers along the Iberian – Newfoundland margins and the use
of these features for plate kinematic modelling. In addition, we have illustrated the uncertainties in current plate models by
restoring seismic transects to their pre-breakup locations utilising existing rotation schemes of Barremian age. Interpretation
of our new seismic dataset has revealed that:

-   M-series magnetic anomalies are not diagnostic of true oceanic crust beneath the SNB. Instead they are attributed to
    susceptibility contrasts between zones of highly-extended continental crust and exhumed mantle in the basin floor.
    Similarly, the high-amplitude J Anomaly coincides with a zone of exhumed mantle punctuated by significant
volcanic additions, and at times characterised by interbedded volcanics and sediments.

-   In the southern part of the Newfoundland margin, we suggest J-anomaly source bodies to be the result of mantle
    dynamics preceding plate rupture. Previously-published studies show that, further north, the J-anomaly is either too
    weak to recognise, or missing altogether.  Although associated with events immediately preceding first seafloor
    spreading, these events are neither instantaneous in time nor isochronous along the margin, which renders the J
Anomaly unsuitable as a kinematic marker.

-   Our results show that magmatic activity was underway in the SNB at a time coinciding with M4 (128 Ma), earlier
    than previously thought. SDR packages onlapping onto a basement fault suggest that, at this time, plate divergence
    was still being accommodated, at least partially, by tectonic faulting.

-   Differences in the relative positions of Iberia and Newfoundland according to published Barremian age plate
reconstructions built on the basis of structural data vs. magnetic data illustrate the uncertainties introduced into the
    modelling procedure by the use of extended continental margin data (dubious magnetic anomaly identifications,
    breakup unconformity interpretations). In the SNB, we interpret the extent of the COTZ to reach oceanward to at
    least M0 (118 Ma). As a result, a complementary approach is needed for constraining plate kinematics of the
    Iberian plate pre M0 times. In this respect we anticipate the palaeoposition of Iberia could come to be more
confidently reconstructed using a larger more comprehensive plate model that encompasses the central and southern
    North Atlantic Oceans.





- Our new data and previous studies of the Iberia-Newfoundland margins illustrate a diversity of features that define conjugate asymmetry and along-strike variability to the extent that it becomes impossible to unequivocally identify conjugate pairs of seismic transects from their geometric and stratigraphic characteristics alone. Although our new data do not provide sufficient clarity about conjugate pairs of, they are helpful to clarify the temporal context for future plate kinematic reconstructions.

- A robust plate kinematic model built from well-constrained spreading data and involving a larger plate circuit would provide the basis to generate virtual rift-spanning seismic transects at the time of continental break-up. This, in turn, would make it possible to investigate further how the processes related to continental breakup are recorded in the sedimentary architecture of rifted margins. Such a plate model does not yet exist.




**Figures**

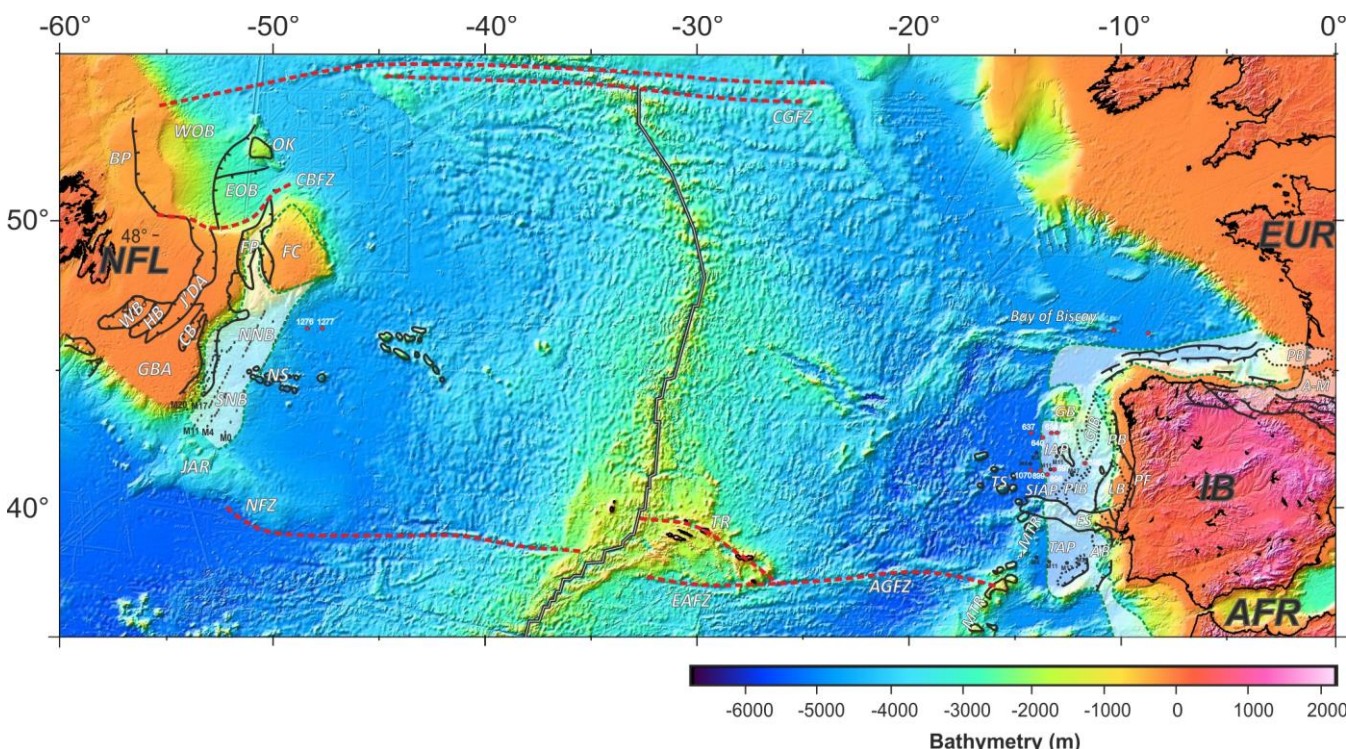

Fig. 1: Study area showing the location of structural and tectonic features significant to our study. White envelopes mark the extent of the COTZ as compiled by Eagles et al., (2015). Magnetic picks as interpreted by Srivastava et al., (2000). Double
black line: mid-ocean ridge; Red dashed lines: fracture zone traces. Background image is derived from Sandwell and Smith (2014) gridded satellite-derived bathymetry using the Generic Mapping Tool, Wessel & Luis, (2017). AB, Alentejo Basin; AM, Amorican Margin; BP, Bonavista Platform; CB; Carson Basin, CBFZ; Cumberland Belt Transfer Zone, EOB; East Orphan Basin, ES; Estremadura Spur, FC; Flemish Cap, FP; Flemish Pass, GB; Galicia Bank, GBA; Grand Banks, GIB; Galicia Interior Basin, HB; Horseshoe Basin, IAP; Iberian Abyssal Plain, IB; Iberia, J'DA; Jeanne d'Arc Basin, LB;
Lustanian Basin, MTR; Madeira-Tore Rise, NFL; Newfoundland, NS; Newfoundland Seamounts, NNB; North Newfoundland Basin, NB; Southern Newfoundland Basin, OK; Orphan Knoll, PB; Parentis Basin, POB; Porto Basin PIB; Peniche Basin, SIAP; Southern Iberian Abyssal Plain, TAP; Tagus Abyssal Plain, TS; Tore Seamounts, WB; Whale Basin, WOB; West Orphan Basin.








Fig. 2: Six stages of development of the North Atlantic, from Late Jurassic to Late Cretaceous. Bright green envelopes show the maximum extent of the Continent-Ocean Transition Zone (Pérez-Diaz and Eagles, 2014). Light green shading shows




oceanic lithosphere extent according to Sibuet et al., (2007) for the Atlantic and Sibuet, et al., (2004) for the Bay of Biscay. Adapted from Vissers and Meijer, (2012)

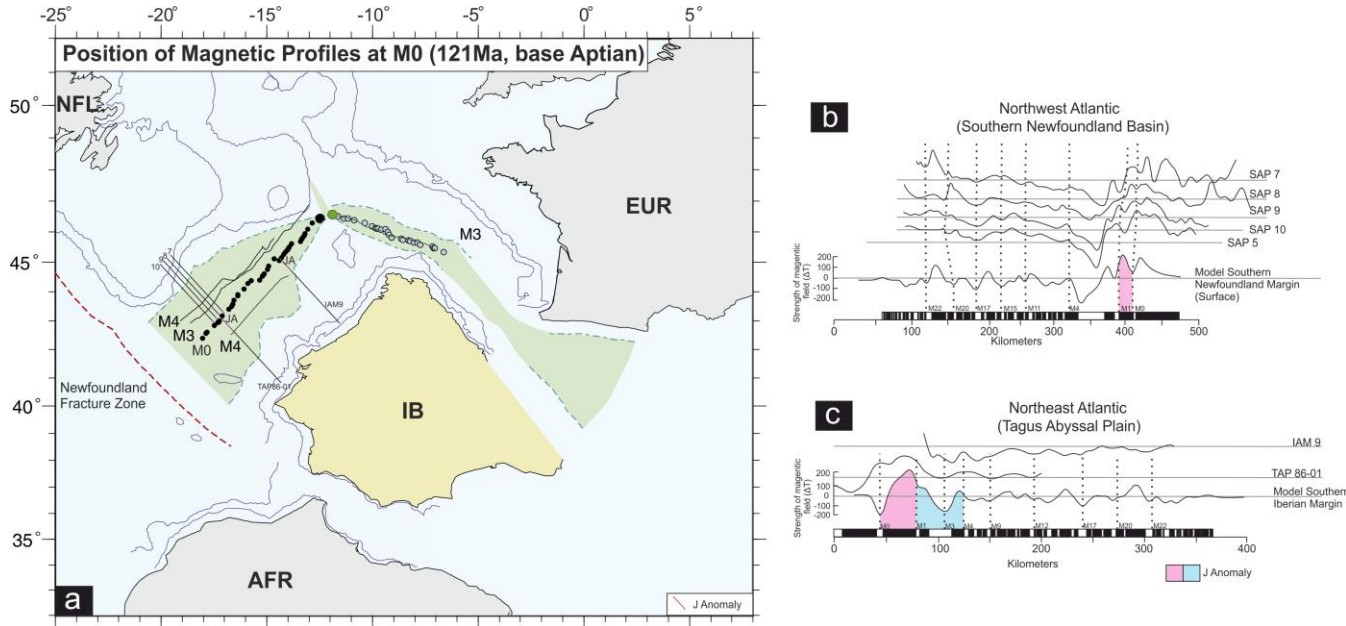

Fig. 3: Magnetic profiles taken across the Southern Newfoundland Basin and conjugate Tagus Abyssal Plain. The J Anomaly corresponds to the high amplitude portion of the profiles, identified as M0 - M1 by Rabinowitz et al., (1978) shown in pink, and M0 – M4 in the Tagus Abyssal Plain (Whitmarsh and Miles, 1995) shown in blue. Profiles have been adapted from Srivastava et al., (2000).





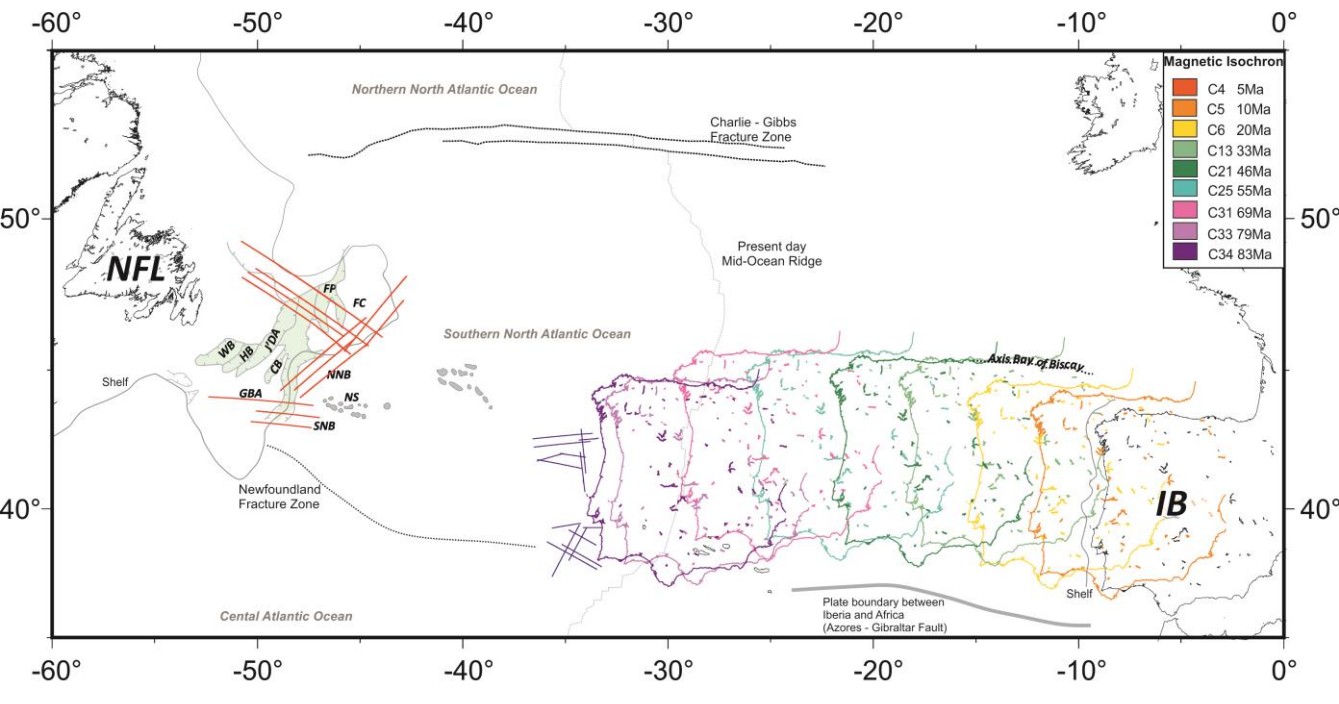

Fig. 4: Map showing the positions of Iberia, relative to North America from first unequivocal oceanic crust (83Ma). Blue and red lines are the TGS Iberian and Newfoundland datasets, respectively. Positions of seismic lines were provided by TGS. Abbreviations as figure 1.













Fig. 5: Interpreted seismic reflection profile ("Line A") from the southern South Newfoundland Basin, offshore Newfoundland. Interpretation shows the basement structure and sedimentary units. (a) Basement structure at the base of the continental slope, (b) Ocean-ward dipping reflectors in the syn-rift 1 sediments, shows fault migration ocean-ward, (c) Volcanic edifice present in the proto-oceanic zone with associated sills and magmatic vents. All data courtesy of TGS.

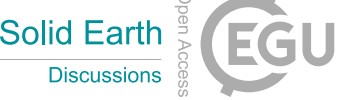

Fig. 6: Interpreted seismic reflection profile ("Line B") from the central South Newfoundland Basin, offshore Newfoundland. Interpretation shows the basement structure and sedimentary units. (a) Crustal collapse of the hanging-wall in a large scale landward dipping fault within extremely thinned continental crust, (b) Section of syn-rift sediments within the exhumed mantle zone, shown to be rotated toward the continent, (c) Bright amplitude reflectors which dip oceanward, a mixture of sediment and magmatic flows beneath an igneous top basement. All data courtesy of TGS.





Fig. 7: Interpreted seismic reflection profile ("Line C") from the northern South Newfoundland Basin, offshore Newfoundland. Interpretation shows the basement structure and sedimentary units. (a) Continental crust thinned by small normal faults, (b) Possible serpentinite diaper within the basement high of the zone of exhumed mantle. All data courtesy of TGS.





695

Fig. 8: Reconstructions of the COTZ envelope from Eagles et al. (2015) at (a) Aptian, (b) Barremian and (c) Tithonian ('full fit) times, showing the range of virtual conjugates generated by alternative rotation schemes. Blue and red lines are the TGS Iberian and Newfoundland datasets, respectively. The positions of lines were provided by TGS. See figure 4 for abbreviations.

700





705

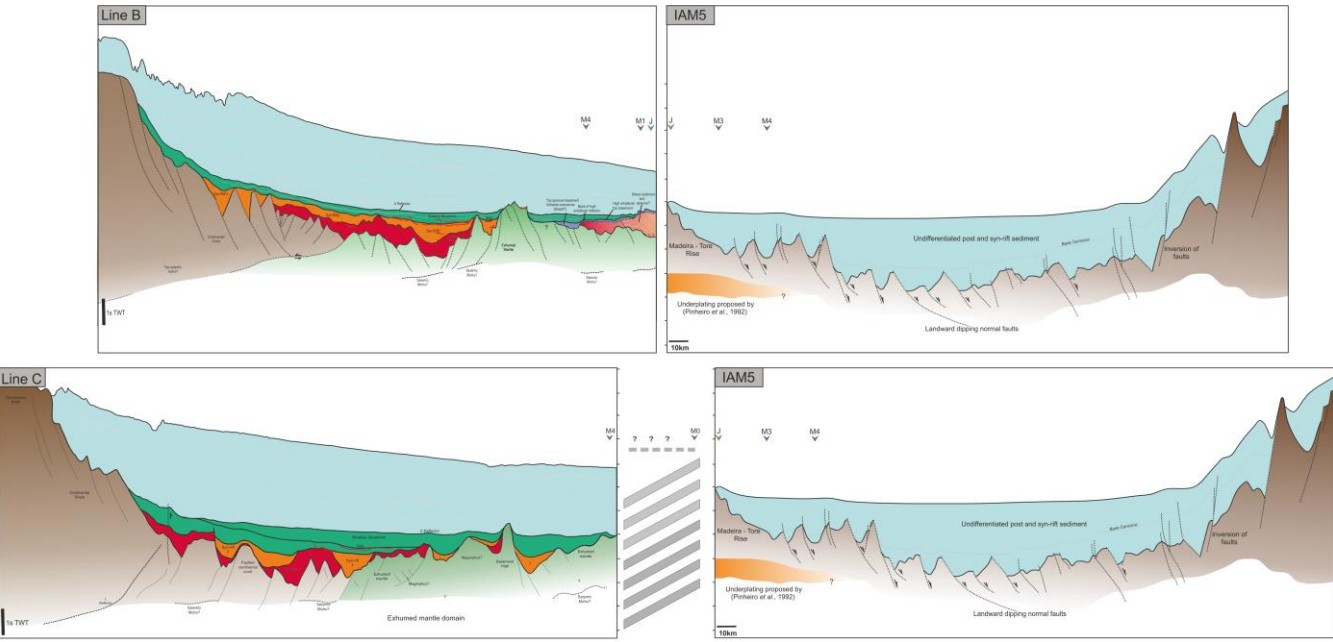

710

Fig. 9: Comparison of 'conjugate' seismic lines chosen on the basis of alternative rotation schemes for Barremian times. (a) Conjugates according to Greiner and Neugebauer, (2013) and Seton et al., (2013) and (b) Srivastava et al., (1990). Conjugate comparisons are hung on 10s TWT. Newfoundland data are courtesy of TGS. Key as in figures 6-8.

715





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
