# Peer review of "Uncertainties in breakup markers along the Iberia-Newfoundland margins illustrated by new seismic data"

_Solid Earth, 2019_

## Referee Comment (RC1) · Alexander L. Peace (Referee) · 11 Oct 2019

In their paper "Uncertainties in breakup markers along the Iberia-Newfoundland margins illustrated by new seismic data" Causer et al. use seismic data from offshore Newfoundland to assess the suitability of commonly used break-up markers along the Newfoundland margin for plate kinematic reconstructions. According to their results, basement associated with the younger M-Series magnetic anomalies is comprised of exhumed mantle and magmatic additions, and therefore most likely represents transitional domains rather than true oceanic lithosphere. This seems reasonable although some aspects of this are hard to assess with the materials currently provided with the

manuscript. This has implications for plate tectonic modelling which is well demonstrated in the paper. The paper is on a worthwhile subject, and Solid Earth seems like an appropriate location for the results of this study. Plate reconstructions in the southern North Atlantic have been the focus of a number of recent publications, demonstrating that this is a very topical subject (Barnett-Moore et al., 2018; Nirrengarten et al., 2018; Peace et al., 2019). In addition, although the Newfoundland-Iberia margins are one of the most studied conjugate margin pairs in the world, there remains significant unknowns regarding the early aspects of separation (Eddy et al., 2017). Thus, the topic of the study addresses a very relevant subject.

Overall, the study seems to be generally well thought out and suitable for publication. However, there are several aspects that I think could be drastically improved, as outlined in detail below. I would therefore like to offer a largely supportive review on this paper, with a recommendation that this paper is published following major revisions.

1) Applying results beyond the data coverage

It is reasonable to extrapolate the finding of the study somewhat beyond the area investigated. However, further consideration, and justification, of how feasible this is would substantially improve the manuscript. Specifically, limited 2D seismic reflection data is interpreted on the continental margins and this is used to derive implications for plate models of the entire region. Although I think the approach is probably valid, it could potentially be problematic because it is well established that passive continental margins are highly variable along strike, so observations made in a region are not necessarily applicable elsewhere without consideration of the processes involved. For example, breakup of the southern North Atlantic occurred via a propagating rift (e.g., Nirrengarten et al., 2018), so timing of rifting and breakup is not the same right along the margin, and also the margin is highly structurally variable, with local complexities such as magmatism and reactivation. As such, the interpretation of magnetic anomalies source using the limited seismic data may not be valid for the entire anomaly. The authors should consider this aspect further in their justification of the approach, and
also in the subsequent discussion section.

2) Location and orientation of the lines

The location and orientation of Lines A-C is currently difficult to discern with the current figure setup and description in the manuscript. For example, although the complete seismic grid is shown (Fig. 4), none of the figures show which line within the grid is Line A-C. As such, it is problematic to fully assess the validity of the results and outcomes. This links with the issue outlined above regarding the validity of the results over the entire region. This could in part be rectified by addressing the issues with the figures outlined below. In addition, although a sparse grid of 2D lines is shown on some of the figures only three lines are presented in detail in the paper. It would be beneficial if the authors could provide further description of what else is shown by the other lines in the grid of seismic data, and also describe why they have chosen lines A-C over others. Finally, the nature of the blue seismic grid shown on the Iberian margin is not well described in the manuscript.

3) Deformable models

The fundamental subject of the paper is about how current plate kinematic models of the Newfoundland-Iberia conjugate margins do not sufficiently describe the separation, and lead to problems when reconciled with regional observations. This aspect is well outlined in the paper. Recent work however, has sought a new solution to this issue through the use of deformable plate tectonic modelling, to reduce overlap in reconstructed conjugate margins and develop concepts of plate kinematics (Ady and Whittaker, 2018; Müller et al., 2019; Peace et al., 2019). These models are far from perfect but offer an alternative approach to the problem addressed in the paper. I think that discussion of the role of this new approach to plate modelling would also be beneficial in the manuscript.

4) Figures

In my opinion the figures are currently one of the weakest aspects of the manuscript. Overall, I felt that they were: 1) underutilised in the text, 2) difficult to interpret, and 3) at times ambiguous. Generally, on all figures making the text larger would substantially improve them. As outlined in the points below, the figures need substantial work to be of publication quality. In addition, I think adding an new figure showing a magnetic anomaly map of the region as a new Figure 2 would substantially improve the manuscript. This would be very beneficial to those working outside of the present study area as it could be used to label feature such as the J-Anomaly and M-Series. Something like the EMAG model (Maus et al., 2009) would suffice here.

Figure 1: I felt that figure 1 could have been used much more extensively throughout the manuscript. In particular, I think it could be used to show the locations of the other figures, and the data, as well as providing a better description of the geological setting such as the key magnetic anomalies. Also, many aspects of this figure are very problematic to see and interpret. For example, the red dots indicating drill sites are nearly impossible to find. In addition, although many of these are referred to in the text (e.g., DSDP site 398) there appear to be some wells in the Bay of Biscay without labels leaving me wondering what is the relevance of these? The green dashed line is not defined in the caption, and the "white envelopes" are difficult to see. Moreover, the red dashed lines do not show all the oceanic fracture zones, so why have these ones been chosen specifically?

Figure 2: Text is again too small. In addition, what is the small circle within 'the maximum extent of the Continent-Ocean Transition Zone' at 83 Ma offshore Newfoundland (under the 'B' of 'Base').

Figure 3: It is not immediately clear to the reader where the magnetic profiles shown in parts b and c are located. In addition, the text is again too small. Finally, what are the black dots shown on a, they are not described in the legend.

Figure 4: This figure is integral to the study as it shows the location of the data. However, it is difficult to know which line presented in the paper (i.e. Lines A-C) corresponds which location shown on the figure. This information needs adding to the figure, otherwise the reader is unable to locate the data. Also the age of the isochrons quoted on the figure are according to which timescale?

Figure 5-7: Although the general interpretations shown look reasonable, there are several aspects of these figures that need substantial improvement. First, the labelling of subfigures (a-c) on these figures is a little strange as the seismic line and its interpretation are not given a subfigure letter. Another thing that struck me when I first saw the interpreted sections was that ages are provided for the sediment packages (e.g., Late Cretaceous), yet in the text it is stated that "sediments have been grouped into Synrift 1, Synrift 2, Breakup-sequence, and Post-Rift packages based on seismicstratigraphic observations". Given this, where have these ages come from? In addition, it would help if the scale bars for TWT and distance were also present on the seismic data. Also, on some of the figures sills are labelled, how are these differentiated from other high amplitude reflectors? Finally, the difference between the grey and the black lines (in the key) is impossible to determine on the figure, the L-reflector symbol is too similar to the fault symbol, and the text on all of these figures is too small.

Figure 8: I like the approach to showing reconstruction using different models, however the text on this figure is again too small, particularly the age in Ma.

Figure 9: I think the concept behind figure 9 is good, particularly the description in the text acknowledging the limitations in this approach. However, all of the text on this figure either needs to be made substantially larger or removed. If all of the interpretation has been shown previously perhaps the text can be removed from this figure.

5) References

Throughout the manuscript there are multiple statements that require references. In. particular, when the 'literature' is referred to or a statement like 'broadly accepted' is used, I think it is necessary to add additional references. Specific examples are of this

are provided below. In addition, a few references are cited in the paper that do not occur in the reference list. For example, Eagles et al. (2015) is not in the reference list. Furthermore, the citation of 'in prep' works seems unnecessary given that the statement being supported could be supported with other published works. For example, at line 235 the compilation model of Matthews et al. (2016) could be cited as this also includes independent plates for Newfoundland (as part of North America), Iberia, Eurasia and Africa, as do other models (e.g., Nirrengarten et al., 2018). In addition, in plate modelling one can keep adding more and more plates, building increasingly complex models so what would be different about the model cited as 'in prep'? For example, Nirrengarten et al. (2018) use independent plates (with separate poles) for Flemish Cap, Rockall-Hatton Bank, Orphan Knoll and also parts of Iberia. Perhaps, this aspect is worthy of discussion in the paper.

Minor points:

Line 14: I suggest replacing 'on the belief' with another phrase such as 'based on the concept'.

Lines 14-15: What exactly differs between the models? The timing or the rotations? Inclusion of different plates? Essentially I found this statement a bit vague.

Line 23: I suggest replacing 'to' with 'with' after 'associated'.

Lines 34-38 (opening paragraph of introduction): All the statements in this paragraph need referencing.

Line 45: 'computer generated plate reconstructions' – I found this statement to be quite vague, surely most modern plate reconstructions are done on a computer?

Lines 48-49: 'alternative scenarios proposed in the literature' - Which alternative scenarios, and in what literature? This statements needs references and further description. I know this is described later on but I felt that without references here the statement feels out of place.

Line 51: 'overlaps' – deformable plate modelling goes someway to address this, and I think it would be good to discuss this aspect of plate modelling (Ady and Whittaker, 2018; Müller et al., 2019; Peace et al., 2019).

Line 56: Why say 'West" here but nowhere else when referring to Iberia?

Line 66: 'heavily debated' – By who? This statement needs references, and explanation of what exactly is debatable about the aspects described in the sentence.

Lines 56-75: I felt that this was a really good description of the history and problems associated with studying the Newfoundland Iberia conjugate margins.

Lines 83-84: Slightly awkward phrasing.

Line 85: 'said studies' – which 'said studies'? You should cite them here.

Line 85: 'published rotation schemes'. Again, I think you should say which rotation schemes by citing the appropriate literature.

Line 89: Awkward phrasing. I suggest modifying this.

Line 90: Should the references be in chronological order in Solid Earth papers?

Line 94: Eagles et al. (2015) is not in the reference list.

Line 95: 'gradual' - Is it really gradual? I am just not sure that this is the best description. It is wide and structurally complex, but I don't think we can describe a change in crustal affinity as gradual.

Line 98: 'so-called' - according to whom? Add appropriate references here.

Line 99: I suggest inserting 'the' before 'literature.

Line 100: Which 'literature' is being referred to in the sentence ending here. Add appropriate references.

Lines 101-112 (whole paragraph): I think this paragraph could be summarised to make

it a bit simpler.

Line 108: 'age of seafloor spreading' - Eddy et al. (2017) discuss this. Also, this reference should probably be included generally as its quite recent and integral to the topic.

Lines 114-115: Add appropriate references regarding the complexity of reconstructing the kinematics of the Iberian plate.

Line 120: 'broadly accepted' – By who? Add references.

Line 127: I don't think the italics on the citation are necessary.

Lines 131-132: Cadenas et al. (2018) also conducted a recent study on compression along this boundary that might be of use. Also, the models in Peace et al. (2019) show this compression, and actually overestimate the extent and magnitude of thickening (based on published constraints) implying that the published models do not account well for Iberia's kinematics.

Line 133: I am not sure the italics on the citation are necessary here (and elsewhere).

Line 155: 'generally accepted' – this needs references to show who it is accepted by.

Line 164: Why are these references not at the end of the sentence? As it stands, it is confusing which statement the references are referring to.

Line 167: 'contradictory geological evidence' – you should expand on what this evidence is and provide references.

Line 167: 'Site 1070' – This is very difficult to see on figure 1.

Line 178: "old oceanic lithosphere' – How old? If you can provide an age here it would be better.

Line 186: 'The J-Anomaly' – See notes in section above regarding a figure showing the magnetic anomaly locations.

Line 196-200: Some references are in italics whilst others are not?

Line 201-208: Same as previous comment regarding italics.

Line 219-225: I found the tense of this paragraph quite strange. Essentially you are describing what you will do so why write it like this?

Line 235: The citation of 'in prep' works seems unnecessary given that the statement being supported could be supported with other published works. This point is expanded on in the points above.

Line 237: Remove 'some'.

Line 252-253: 'sediments have been grouped into Synrift 1, Synrift 2, Breakup-sequence, and Post-Rift packages based on seismic stratigraphic observations' - This statement appears to contradict what is shown on the figures as on the figures the sediments are also given ages? Also, where have these ages come from? I suggest providing the source of the information.

Line 263: 'DSDP, Site 298' - This is very hard to see on Figure 1. I suggest making this larger, along with all the other wells shown on the figure.

Line 274: 'variable offsets' – This is quite a vague phrase. Can these offsets be quantified on the data?

Line 275: 'seismic Moho' – refer to the figure showing this?

Line 289: 'Fig 5c' - This is good, I suggest referring to the subfigures more often when describing the interpretation.

Line 295: 'distorted seismic imaging' - This is quite vague terminology.

Line 299: Again, which line on the figure showing the seismic grid is line B?

Line 320: As with previous comment but for Line C.

Line 350 onwards (opening paragraph of the Discussion): I found the whole of this first

paragraph of the discussion to be very vague, and question whether it is fully necessary as much of this information has already been provided in the introductory sections.

Line 350: 'three seismic lines' – why is a grid of seismic lines shown but only three are presented in the paper? Did you analyses the others, and how did you choose the ones presented?

Line 368: Yamasaki and Gernigon (2009) do not mention the origin of SDRs in their paper, so this citation does not make sense here.

Line 406-408: Opening statement on conjugate margins - This is good, I like that you state this.

References cited:

Ady, B.E., and Whittaker, R.C., 2018, Examining the influence of tectonic inheritance on the evolution of the North Atlantic using a palinspastic deformable plate reconstruction: Geological Society of London, Special Publications, v. 470, doi:10.1144/SP470.9.

Barnett-Moore, N., Müller, R.D., Williams, S., Skogseid, J., and Seton, M., 2018, A reconstruction of the North Atlantic since the earliest Jurassic: Basin Research, v. 30, p. 160–185, doi:10.1111/bre.12214.

Cadenas, P., Fernández-Viejo, G., Pulgar, J.A., Tugend, J., Manatschal, G., and Minshull, T.A., 2018, Constraints Imposed by Rift Inheritance on the Compressional Reactivation of a Hyperextended Margin: Mapping Rift Domains in the North Iberian Margin and in the Cantabrian Mountains: Tectonics, v. 37, p. 758–785, doi:10.1002/2016TC004454.

Eagles, G., Pérez-Díaz, L., and Scarselli, N., 2015, Getting over continent ocean boundaries: Earth-Science Reviews, v. 151, p. 244–265, doi:10.1016/j.earscirev.2015.10.009.

Eddy, M.P., Jagoutz, O., and Ibañez-Mejia, M., 2017, Timing of initial seafloor spreading

in the Newfoundland-Iberia rift: Geology, v. 45, p. G38766.1, doi:10.1130/G38766.1.

Matthews, K.J., Maloney, K.T., Zahirovic, S., Williams, S.E., Seton, M., and Müller, R.D., 2016, Global plate boundary evolution and kinematics since the late Paleozoic: Global and Planetary Change, v. 146, p. 226–250, doi:10.1016/j.gloplacha.2016.10.002.

Maus, S. et al., 2009, EMAG2: A 2-arc min resolution Earth Magnetic Anomaly Grid compiled from satellite, airborne, and marine magnetic measurements: Geochemistry, Geophysics, Geosystems, v. 10, doi:10.1029/2009GC002471.

Müller, R.D. et al., 2019, A global plate model including lithospheric deformation along major rifts and orogens since the Triassic: Tectonics, doi:10.1029/2018TC005462.

Nirrengarten, M., Manatschal, G., Tugend, J., Kusznir, N., and Sauter, D., 2018, Kinematic evolution of the southern North Atlantic: implications for the formation of hyper-extended rift systems: Tectonics, p. 2, doi:10.1002/2017TC004495.

Peace, A.L., Welford, J.K., Ball, P.J., and Nirrengarten, M., 2019, Deformable plate tectonic models of the southern North Atlantic: Journal of Geodynamics, doi:10.1016/j.jog.2019.05.005.

Yamasaki, T., and Gernigon, L., 2009, Styles of lithospheric extension controlled by underplated mafic bodies: Tectonophysics, v. 468, p. 169–184, doi:10.1016/j.tecto.2008.04.024.

---

## Referee Comment (RC2) · Frauke Klingelhoefer (Referee) · 21 Nov 2019

The mansucript "Uncertainties in breakup markers along the Iberia-Newfoundland margins illustrated by new seismic data" by Annabel Causer, Lucía Pérez-Díaz, Jürgen Adam and Graeme Eagles present unpublished seismic data from the Southern Newfoundland Basin to study the impact of commonly used break-up markers for plate cinematic reconstructions of the initial ocean opening between the West Iberia and Newfoundland margins. The main conclusion is that in this region the "traditional" break-up markers do not allow to unequivocally discriminate the validity of the different plate tectonic rotational poles proposed in literature. From this the authors propose

that new and better constrained reconstruction are needed to identify individual seismic profiles as parts of conjugate pairs.

It is a bit unsatisfactorly that the main conclusion of this manuscript is that it is not possible to better constrain the opening using the data presented. A better constraint on the error of the different reconstructions could probably be done using the work of Hellinger, 1981 or Chang, Royer et al., 1991. A tool using these approaches is available in the free Gplates software (https://www.gplates.org/user-manual/HellingerTool.html).

In my opinion, the manuscript is missing some information. It would be nice to know which software has been used for the plate cinematic reconstructions and for data processing. A short description of the seismis data processing, even if done by TGS would be of interest.

The discussion should be extended to give at least an impression of comparable margins. Is this uncertainty a general problem or only in this specific region, which has nonetheless been very extensively investigated ? If only here, than why, for example are the magnetic anomalies especially unclear and uninterpretable or is this due to the large extend of serpentinised mantle material ?

The manuscript has no acknowledgement section, but probably some free software ("Generic mapping tools" or other) were used and should be acknowledged.

Figures: Figure 1: it would be nice to add the magnetc anomaly positions. Figs 5, 6, 7: all panels should be annotted a,b,c,d,e and explained in the legend. I think a classical offset and time annotation would be helpful, rather than just having a scale for one second and 10km. Middle panel have no indication for 0 s. Figure 9: strictly spreaking there are no data shown in this figure, but mentioned in the caption.

Some smaller corrections : L. 82 Furthermore -> Furthermore we L. 94 missing ")" L. 104 "(" too much L. 169 Isn't M25 125 Ma age ? L. 177 "(" too much L. 219-228 This is more "objectivs" than "Data and methods" L. 229 allows -> allow L. 239 Would be

nice to have more detail, seize of the airgun array, length of the streamer... L. 390
suggested -> suggest

Gurnis, M., M. Turner, S. Zahirovic, L. DiCaprio, S. Spasojevic, R. D. Müller, J. Boyden,
M. Seton, V. C. Manea, and D. J. Bower, Plate tectonic reconstructions with contin-
uously closing plates, Computers & Geosciences, 38, 35-42, 2012. Hellinger, S. J.
(1981). The uncertainties of finite rotations in plate tectonics. Journal of Geophysical
Research: Solid Earth, 86(B10), 9312-9318. Chang, T., Ko, D., Royer, J. Y., & Lu, J.
(2000). Regression techniques in plate tectonics. Statistical Science, 342-356.

---

## Referee Comment (RC3) · Anonymous Referee #3 · 5 Dec 2019

General comments: The aim of the manuscript submitted by Causer et al. is to discuss breakup markers along the Iberia-Newfoundland margins based on new seismic data. The theme of the manuscript is of major scientific interest, since neither the nature, not the timing and location of breakup are well constrained along the Iberia-Newfoundland margins. Many papers, some of which are very recent, have been dedicated to this problem. I have to admit that I did not find new ideas, or new, well constrained observations that add something new to the subject. Indeed, the interpretation of the new seismic data lack a rigorous interpretation and observations and interpretations are mixed and difficult to follow (for some further comments see comments below). The manuscript reads more like a report referring to old studies and only very few new ob-

servations are added. Most disturbing is that some of the latest studies, that come to almost the same conclusions, are only marginally referred or partly not discussed. This omission weighs heavily and discredit the authors. Apart from these points, there are several other points (see comments below) that makes that this manuscript can not be accepted in its present version.

Specific comments: l.30: here and elsewhere in the paper the authors make statements that are similar to the papers of Nirrengarten et al, without citing their work. Actually, most of the conclusions reached in this paper are similar to those by Nirrengarten et al. 2017 and 2018, thus, referring to these results is necessary. I would propose that the authors should discuss how their results are different from those of Nirrengarten et al. 2017 and 2018. I do not really see a big difference. Moreover, the papers of Stanton et al. 2015 and Nirrengarten et al. 2018 that deal with the same subject are not referred to. l. 25: the SDR packages need to be better described; what is the origin (magmatic) and the significance of the SDR package? I can not find them in the figures l.130 to 150: the tectonic setting part is completely outdated. A lot of work has been done in the last years that need to be referred to. l.218 Dataset and methods section need to be rewritten and more details about the data presented in the paper need to be presented. l.265 to 345 The description of the seismic lines mixes observations with interpretations. Many questions remain open, such as how syn-rift 1 and 2 have been defined, where are possible limits, how were different types of basement defined and what are the evidence for magmatic additions (there are many more questions that arise by looking the seismic interpretations). The presentation of the data needs to include the presentation of the seismic section (without interpretation), a line drawing and the presentation. As presented here, I cannot follow the interpretations and many of the assumption are not back up by observations. The presentation of the seismic data is insufficient and does not corresponds to the standard of scientific papers. l.360 to 400: this section does not really discuss new ideas and does not built on the observations neither. Most of what is said here is old and outdated (the authors seem to have missed the research on the Iberia-Newfoundland margins of the last 5

years??) l.405 to 460: This section reads more as a report than a discussion chapter.

---

## Author Comment (AC1) · 18 Dec 2019

**Uncertainties in breakup markers along the Iberia-Newfoundland margins illustrated by new seismic data**

Annabel Causer[1], Lucía Pérez-Díaz[1,2], Jürgen Adam[1] and Graeme Eagles[3]

[1]Earth Sciences Department, Royal Holloway University of London, Egham, TW20 0EX, United Kingdom
[2]Department of Earth Sciences, Oxford University, Oxford, OX1 3AN, United Kingdom
[3]Alfred Wegener Institut, Helmholtz Zentrum für Polar und Meeresforschung, Bremerhaven, Germany

Correspondence to: Annabel Causer (annabel.causer.2017@live.rhul.ac.uk)

**Response to reviews**

**Reviewer 1: Alexander Peace**

In their paper "Uncertainties in breakup markers along the Iberia-Newfoundland margins illustrated by new seismic data" Causer et al. use seismic data from offshore Newfoundland to assess the suitability of commonly used break-up markers along the Newfoundland margin for plate kinematic reconstructions. According to their results, basement associated with the younger M-Series magnetic anomalies is comprised of exhumed mantle and magmatic additions, and therefore most likely represents transitional domains rather than true oceanic lithosphere. This seems reasonable although some aspects of this are hard to assess with the materials currently provided with the manuscript. This has implications for plate tectonic modelling which is well demonstrated in the paper.

The paper is on a worthwhile subject, and Solid Earth seems like an appropriate location for the results of this study. Plate reconstructions in the southern North Atlantic have been the focus of a number of recent publications, demonstrating that this is a very topical subject (Barnett-Moore et al., 2018; Nirrengarten et al., 2018; Peace et al., 2019). In addition, although the Newfoundland-Iberia margins are one of the most studied conjugate margin pairs in the world, there remains significant unknowns regarding the early aspects of separation (Eddy et al., 2017). Thus, the topic of the study addresses a very relevant subject.

Overall, the study seems to be generally well thought out and suitable for publication. However, there are several aspects that I think could be drastically improved, as outlined in detail below. I would therefore like to offer a largely supportive review on this paper, with a recommendation that this paper is published following major revisions.

*No action: We thank Alexander Peace for his fair and constructive review of our work. Below, we outline the changes we've made to our manuscript in light of his comments. References to line numbers are made with respect to the revised version of the manuscript.*

**1) Applying results beyond the data coverage**

It is reasonable to extrapolate the finding of the study somewhat beyond the area investigated. However, further consideration, and justification, of how feasible this is would substantially improve the manuscript. Specifically, limited 2D seismic reflection data is interpreted on the

continental margins and this is used to derive implications for plate models of the entire region. Although I think the approach is probably valid, it could potentially be problematic because it is well established that passive continental margins are highly variable along strike, so observations made in a region are not necessarily applicable elsewhere without consideration of the processes involved. For example, breakup of the southern North Atlantic occurred via a propagating rift (e.g., Nirrengarten et al., 2018), so timing of rifting and breakup is not the same right along the margin, and also the margin is highly structurally variable, with local complexities such as magmatism and reactivation. As such, the interpretation of magnetic anomalies source using the limited seismic data may not be valid for the entire anomaly. The authors should consider this aspect further in their justification of the approach, and also in the subsequent discussion section.

*Action: We fully agree with the reviewer and have made this clearer in our revised manuscript (lines 97-99)*

**2) Location and orientation of the lines**

The location and orientation of Lines A-C is currently difficult to discern with the current figure setup and description in the manuscript. For example, although the complete seismic grid is shown (Fig. 4), none of the figures show which line within the grid is Line A-C. As such, it is problematic to fully assess the validity of the results and outcomes.

*Action: We have added Lines A-C to figures 1 and 4, and made reference to them in the text where appropriate.*

This links with the issue outlined above regarding the validity of the results over the entire region. This could in part be rectified by addressing the issues with the figures outlined below. In addition, although a sparse grid of 2D lines is shown on some of the figures only three lines are presented in detail in the paper. It would be beneficial if the authors could provide further description of what else is shown by the other lines in the grid of seismic data, and also describe why they have chosen lines A-C over others. Finally, the nature of the blue seismic grid shown on the Iberian margin is not well described in the manuscript.

*Action: The grids of lines have been removed from figure 4, and lines A-C are shown instead. We have maintained the grid on figure 8 as we believe this helps the reader better understand the implications of choosing conjugates on the basis of alternative plate models. The revised text includes a statement that we chose to present lines A-C because they cover the overall range of possible locations for the conjugate to IAM-5.*

**3) Deformable models**

The fundamental subject of the paper is about how current plate kinematic models of the Newfoundland-Iberia conjugate margins do not sufficiently describe the separation, and lead to problems when reconciled with regional observations. This aspect is well outlined in the paper. Recent work however, has sought a new solution to this issue through the use of deformable plate tectonic modelling, to reduce overlap in reconstructed conjugate margins and develop concepts of plate kinematics (Ady and Whittaker, 2018; Müller et al., 2019; Peace et al., 2019). These models are far from perfect but offer an alternative approach to the

problem addressed in the paper. I think that discussion of the role of this new approach to plate modelling would also be beneficial in the manuscript.

*Action: We agree that deformable models do present an alternative approach to studying highly extended continental margins, and techniques such as these have been worked on in the past years (e.g. Ady and Whittaker, 2018; Müller et al., 2019; Peace et al., 2019). Deformable models such as these are founded on assumptions which integrate uncertainties investigated in this paper, for example the COB and M-Series. Recent work by Eagles et al., (2015) found that the choice of COB only has a modest effect on its planispatically-restored equivalent, as COB estimates are reduces by stretching factor. As a result statistical uncertainties are greater for deformable models than the more conventional methods of rotating points around a stage pole.*

*We have included the suggested references, and added detail to the text regarding deformable models (Lines 51-65).*

**4) Figures**

In my opinion the figures are currently one of the weakest aspects of the manuscript. Overall, I felt that they were: 1) underutilised in the text, 2) difficult to interpret, and 3) at times ambiguous.

Generally, on all figures making the text larger would substantially improve them.

*Action: Done*

As outlined in the points below, the figures need substantial work to be of publication quality. In addition, I think adding a new figure showing a magnetic anomaly map of the region as a new Figure 2 would substantially improve the manuscript. This would be very beneficial to those working outside of the present study area as it could be used to label feature such as the J-Anomaly and M-Series. Something like the EMAG model (Maus et al., 2009) would suffice here.

*Action: We feel that an extra gridded magnetic anomaly map would take up too much space to justify only for the purpose of locating anomaly J and the disputed M-series isochrons. Instead, we have added the location of the J and M-Series anomalies to figure 1 for reference.*

**Figure 1:** I felt that figure 1 could have been used much more extensively throughout the manuscript. In particular, I think it could be used to show the locations of the other figures, and the data, as well as providing a better description of the geological setting such as the key magnetic anomalies. Also, many aspects of this figure are very problematic to see and interpret. For example, the red dots indicating drill sites are nearly impossible to find. In addition, although many of these are referred to in the text (e.g., DSDP site 398) there appear to be some wells in the Bay of Biscay without labels leaving me wondering what is the relevance of these? The green dashed line is not defined in the caption, and the "white envelopes" are difficult to see. Moreover, the red dashed lines do not show all the oceanic fracture zones, so why have these ones been chosen specifically?

*Action: Done*

***Figure 2:*** Text is again too small. In addition, what is the small circle within 'the maximum extent of the Continent-Ocean Transition Zone' at 83 Ma offshore Newfoundland (under the 'B' of 'Base').

*Action: We have increased text size and removed the small circle which was in this figure by error.*

***Figure 3***: It is not immediately clear to the reader where the magnetic profiles shown in parts b and c are located. In addition, the text is again too small. Finally, what are the black dots shown on a, they are not described in the legend.

*Action: We have increased text size and improved the figure's labelling. The black dots are the locations of picks on the younger (oceanward) edge of anomaly J made on magnetic profiles that are not included in the rest of the figure.*

***Figure 4:*** This figure is integral to the study as it shows the location of the data. However, it is difficult to know which line presented in the paper (i.e. Lines A-C) corresponds which location shown on the figure. This information needs adding to the figure, otherwise the reader is unable to locate the data. Also the age of the isochrons quoted on the figure are according to which timescale?

*Action: We have modified the figure to show the position of lines A-C. We have also clarified the timescale used.*

***Figure 5-7:*** Although the general interpretations shown look reasonable, there are several aspects of these figures that need substantial improvement. First, the labelling of subfigures (a-c) on these figures is a little strange as the seismic line and its interpretation are not given a subfigure letter. Another thing that struck me when I first saw the interpreted sections was that ages are provided for the sediment packages (e.g., Late Cretaceous), yet in the text it is stated that "sediments have been grouped into Synrift 1, Synrift 2, Breakup-sequence, and Post-Rift packages based on seismicstratigraphic observations". Given this, where have these ages come from? In addition, it would help if the scale bars for TWT and distance were also present on the seismic data. Also, on some of the figures sills are labelled, how are these differentiated from other high amplitude reflectors? Finally, the difference between the grey and the black lines (in the key) is impossible to determine on the figure, the L-reflector symbol is too similar to the fault symbol, and the text on all of these figures is too small.

*Action: Seismic lines have been updated and the ages of syn-rift 1 and syn-rift 2 have been removed (they were there from an early iteration of the manuscript). The source for the age of the U reflector has been referenced in the updated text. We have further improved figure quality by:*

- *Using colour-coded symbols: e.g. exhumation/detachment faults are now shown in red; seismic moho is now shown in blue.*
- *Adding scale bars to all.*
- *Increasing the size of distance bars.*

*Our reasoning behind the interpretation of sills is made clear in the revised text (Lines 311-312)*

**Figure 8:** I like the approach to showing reconstruction using different models, however the text on this figure is again too small, particularly the age in Ma.

*Action: done.*

**Figure 9:** I think the concept behind figure 9 is good, particularly the description in the text acknowledging the limitations in this approach. However, all of the text on this figure either needs to be made substantially larger or removed. If all of the interpretation has been shown previously perhaps the text can are provided below.

*Action: We have condensed the text down to the key points and increased size for readability.*

**5) References**

Throughout the manuscript there are multiple statements that require references. In. particular, when the 'literature' is referred to or a statement like 'broadly accepted' is used, I think it is necessary to add additional references. Specific examples are of this are provided below.

*Action: we have added references in light of the reviewer's detailed comments.*

In addition, a few references are cited in the paper that do not occur in the reference list. For example, Eagles et al. (2015) is not in the reference list.

*Action: done*

Furthermore, the citation of 'in prep' works seems unnecessary given that the statement being supported could be supported with other published works. For example, at line 235 the compilation model of Matthews et al. (2016) could be cited as this also includes independent plates for Newfoundland (as part of North America), Iberia, Eurasia and Africa, as do other models (e.g., Nirrengarten et al., 2018). In addition, in plate modelling one can keep adding more and more plates, building increasingly complex models so what would be different about the model cited as 'in prep'? For example, Nirrengarten et al. (2018) use independent plates (with separate poles) for Flemish Cap, Rockall-Hatton Bank, Orphan Knoll and also parts of Iberia. Perhaps, this aspect is worthy of discussion in the paper.

*Action: we have maintained the citation to our work in preparation and have clarified how this on-going work differs from those mentioned by the reviewer here (lines 249-252).*

*By adding more and more small plates bounded by COBs or disputed M-series isochrons, as the reviewer describes and as has been done before, the interpretational uncertainty in breakup markers that is the subject of our manuscript is not only ignored, but potentially also magnified by propagation through rotations in neighbouring branches of the model. As we explain in the revised text, the aim of our work in preparation is not to increase model complexity in this way, but to reduce model uncertainty by interrogating the set of statistically-permissible combinations of a small number of uncontroversial large-plate*

*models with the aim of finding which of the wide range of Iberia-Newfoundland breakup marker interpretations are viable and, of these, which are most likely.*

**5) Minor points:**

Line 14: I suggest replacing 'on the belief' with another phrase such as 'based on the concept'.

*Action: done*

Lines 14-15: What exactly differs between the models? The timing or the rotations? Inclusion of different plates? Essentially I found this statement a bit vague.

*Action: clarified (Line 16)*

Line 23: I suggest replacing 'to' with 'with' after 'associated'.

*Action: done*

Lines 34-38 (opening paragraph of introduction): All the statements in this paragraph need referencing.

*Action: done*

Line 45: 'computer generated plate reconstructions' – I found this statement to be quite vague, surely most modern plate reconstructions are done on a computer?

*Action: changed to "modern".*

Lines 48-49: 'alternative scenarios proposed in the literature' - Which alternative scenarios, and in what literature? This statement needs references and further description. I know this is described later on but I felt that without references here the statement feels out of place.

*Action: References have been added (Lines 53-54).*

Line 51: 'overlaps' – deformable plate modelling goes someway to address this, and I think it would be good to discuss this aspect of plate modelling (Ady and Whittaker, 2018; Müller et al., 2019; Peace et al., 2019).

*Action: done (Lines 58-65).*

Line 56: Why say 'West" here but nowhere else when referring to Iberia?

*Action: rectified.*

Line 66: 'heavily debated' – By who? This statement needs references, and explanation of what exactly is debatable about the aspects described in the sentence.

*No action: We discuss this statement in the same paragraph, immediately after making it (Line 76 onwards)*

Lines 56-75: I felt that this was a really good description of the history and problems associated with studying the Newfoundland Iberia conjugate margins.
*No action*

Lines 83-84: Slightly awkward phrasing.

*Action: Rewritten.*

Line 85: 'said studies' – which 'said studies'? You should cite them here & Line 85: 'published rotation schemes'. Again, I think you should say which rotation schemes by citing the appropriate literature.

*Action: done (Lines 101-105)*

Line 89: Awkward phrasing. I suggest modifying this.

*Action: done*

Line 90: Should the references be in chronological order in Solid Earth papers?

*Action: all references have been changed to chronological order.*

Line 94: Eagles et al. (2015) is not in the reference list.

*Action: done.*

Line 95: 'gradual' - Is it really gradual? I am just not sure that this is the best description. It is wide and structurally complex, but I don't think we can describe a change in crustal affinity as gradual.

*Action: reworded.*

Line 98: 'so-called' - according to whom? Add appropriate references here.

*Action: references added.*

Line 99: I suggest inserting 'the' before 'literature.

*Action: done.*

Line 100: Which 'literature' is being referred to in the sentence ending here. Add appropriate references.

*Action: done.*

Lines 101-112 (whole paragraph): I think this paragraph could be summarised to make it a bit simpler.

*Action: done.*

Line 108: 'age of seafloor spreading' - Eddy et al. (2017) discuss this. Also, this reference should probably be included generally as its quite recent and integral to the topic.
*Action: Reference added.*

Lines 114-115: Add appropriate references regarding the complexity of reconstructing the kinematics of the Iberian plate.

*Action: done.*

Line 120: 'broadly accepted' – By who? Add references.

*Action: done.*

Line 127: I don't think the italics on the citation are necessary.

*Action: done.*

Lines 131-132: Cadenas et al. (2018) also conducted a recent study on compression along this boundary that might be of use. Also, the models in Peace et al. (2019) show this compression, and actually overestimate the extent and magnitude of thickening (based on published constraints) implying that the published models do not account well for Iberia's kinematics.

*Action: done.*

Line 133: I am not sure the italics on the citation are necessary here (and elsewhere).

*Action: done.*

Line 155: 'generally accepted' – this needs references to show who it is accepted by.

*Action: done.*

Line 164: Why are these references not at the end of the sentence? As it stands, it is confusing which statement the references are referring to.

*Action: references moved.*

Line 167: 'contradictory geological evidence' – you should expand on what this evidence is and provide references.

*Action: done.*

Line 167: 'Site 1070' – This is very difficult to see on figure 1.

*Action: Figure 1 has been updated.*

Line 178: "old oceanic lithosphere' – How old? If you can provide an age here it would be better.

*Action: done.*

Line 186: 'The J-Anomaly' – See notes in section above regarding a figure showing the magnetic anomaly locations.

*Action: Added to Figure 1.*

Line 196-200: Some references are in italics whilst others are not?

*Action: rectified.*

Line 201-208: Same as previous comment regarding italics.

*Action: references are now a consistent style.*

Line 219-225: I found the tense of this paragraph quite strange. Essentially you are describing what you will do so why write it like this?

*No action: The style the reviewer refers to conforms to the structure "X would achieve Y, but X is not available at present". We don't see the need to change verbal tense in this instance.*

Line 235: The citation of 'in prep' works seems unnecessary given that the statement being supported could be supported with other published works. This point is expanded on in the points above.

*No Action: see our response to the reviewer's previous mention of this issue.*

Line 237: Remove 'some'.

*Action: done.*

Line 252-253: 'sediments have been grouped into Synrift 1, Synrift 2, Breakupsequence, and Post-Rift packages based on seismic stratigraphic observations' - This statement appears to contradict what is shown on the figures as on the figures the sediments are also given ages? Also, where have these ages come from? I suggest providing the source of the information.

*Action: ages were removed from figures for consistency – they corresponded to tentative ages in an earlier version of the manuscript.*

Line 263: 'DSDP, Site 298' - This is very hard to see on Figure 1. I suggest making this larger, along with all the other wells shown on the figure.

*Action: Figure 1 has been changed.*

Line 274: 'variable offsets' – This is quite a vague phrase. Can these offsets be quantified on the data?

*No action: we don't feel a change here is needed, the reader is referred to the figure.*

Line 275: 'seismic Moho' – refer to the figure showing this?

*Action: done.*

Line 289: 'Fig 5c' - This is good, I suggest referring to the subfigures more often when describing the interpretation.

*Action: done.*

Line 295: 'distorted seismic imaging' - This is quite vague terminology.

*Action: We have described the basis of our interpretation more precisely in the revised text.*

Line 299: Again, which line on the figure showing the seismic grid is line B?

*Action: Lines A-C are now shown on figures 1 and 4 for clarity.*

Line 320: As with previous comment but for Line C.

*Action: Lines A-C are now shown on figures 1 and 4 for clarity.*

Line 350 onwards (opening paragraph of the Discussion): I found the whole of this first paragraph of the discussion to be very vague, and question whether it is fully necessary as much of this information has already been provided in the introductory sections.

*No Action: We believe this paragraph summarises and reminds the reader of the points raised in the results section, and sets the scene for the discussion to follow.*

Line 350: 'three seismic lines' – why is a grid of seismic lines shown but only three are presented in the paper? Did you analyses the others, and how did you choose the ones presented?

*Action: We have added some clarification in the text (Lines 87-91). The three lines presented were chosen on the basis of them 1) being previously unpublished and 2) crossing regions associated with the J and M-series anomalies.*

*We have maintained the grid on figure 8 as we believe that it illustrates how the choice of plate model influences the identification of conjugates.*

Line 368: Yamasaki and Gernigon (2009) do not mention the origin of SDRs in their paper, so this citation does not make sense here.

*Action: Removed.*

Line 406-408: Opening statement on conjugate margins - This is good, I like that you state this.

*No action*

---

## Author Comment (AC2) · 18 Dec 2019

**Uncertainties in breakup markers along the Iberia-Newfoundland margins illustrated by new seismic data**

Annabel Causer[1], Lucía Pérez-Díaz[1,2], Jürgen Adam[1] and Graeme Eagles[3]

[1]Earth Sciences Department, Royal Holloway University of London, Egham, TW20 0EX, United Kingdom
[2]Department of Earth Sciences, Oxford University, Oxford, OX1 3AN, United Kingdom
[3]Alfred Wegener Institut, Helmholtz Zentrum für Polar und Meeresforschung, Bremerhaven, Germany

Correspondence to: Annabel Causer (annabel.causer.2017@live.rhul.ac.uk)

**Response to reviews**

**Reviewer 2: Frauke Klingelhoefer**

The mansucript "Uncertainties in breakup markers along the Iberia-Newfoundland margins illustrated by new seismic data" by Annabel Causer, Lucía Pérez-Díaz, Jürgen Adam and Graeme Eagles present unpublished seismic data from the Southern Newfoundland Basin to study the impact of commonly used break-up markers for plate cinematic reconstructions of the initial ocean opening between the West Iberia and Newfoundland margins. The main conclusion is that in this region the "traditional" break-up markers do not allow to unequivocally discriminate the validity of the different plate tectonic rotational poles proposed in literature.

From this the authors propose:

1) **Major Comments:**

That new and better constrained reconstruction are needed to identify individual seismic profiles as parts of conjugate pairs. It is a bit unsatisfactorly that the main conclusion of this manuscript is that it is not possible to better constrain the opening using the data presented. A better constraint on the error of the different reconstructions could probably be done using the work of Hellinger, 1981 or Chang, Royer et al., 1991. A tool using these approaches is available in the free Gplates software (https://www.gplates.org/user-manual/HellingerTool.html).

*No Action: the reviewer has not appreciated the main aim of our manuscript, which is to use new data to highlight the large degree of uncertainty involved in interpreting breakup features of the kind that are often used to lead quantitative plate reconstructions. These aims are clearly outlined in section 1 (lines 87-105). Unfortunately, Chang's statistical tools are only applicable with Hellinger's fit criterion for seafloor spreading data. Regardless of how available these tools are in GPlates, they would only be applicable for a small subset of the cited plate reconstructions (those that only use seafloor spreading data). These tools are useless for assessing the uncertainty in geological markers like COBs off Iberia and Newfoundland, or transtensional basins in the Pyrenees. We do aim to take a quantitative statistical approach to understanding the study region in future work, based on a suite of purpose-built two-plate models for Africa, North America, Eurasia, Greenland and Iberia using a more modern and robust inversion scheme. This work is still in progress, and well beyond the scope of this manuscript.*

In my opinion, the manuscript is missing some information. It would be nice to know which software has been used for the plate cinematic reconstructions and for data processing. A short description of the seismic data processing, even if done by TGS would be of interest.

*Action: We have added detail on seismic processing to the revised manuscript (lines 254-265). The caption of figure 4 acknowledges plate modelling method used. Given that plate kinematic modelling is not the principal aim of our manuscript we don't see a need to include further details in the text.*

The discussion should be extended to give at least an impression of comparable margins. Is this uncertainty a general problem or only in this specific region, which has nonetheless been very extensively investigated? If only here, than why, for example are the magnetic anomalies especially unclear and uninterpretable or is this due to the large extend of serpentinised mantle material?

*No Action: we refer the reviewer back to our introduction section, in which the difficulties of interpretation at divergent continental margins in general are introduced by citing a previous global study in which some of us were involved. More specifically, as our study region is the type region for mantle exhumation in wide transition zones, we feel there is little to be gained from a detailed examination of comparable margins where the difficulties of interpretation are likely to be understood with reference to Iberia-Newfoundland.*

The manuscript has no acknowledgement section, but probably some free software ("Generic mapping tools" or other) were used and should be acknowledged.

*Action: GMT will be acknowledged in the final manuscript.*

**Figures:**

Figure 1: it would be nice to add the magnetc anomaly positions.

*Action: Done.*

Figs 5, 6, 7: all panels should be annotted a,b,c,d,e and explained in the legend. I think a classical offset and time annotation would be helpful, rather than just having a scale for one second and 10km. Middle panel have no indication for 0 s.

*Action: figures have been improved and re-labelled in response to the comments here and those of Reviewer 1.*

Figure 9: strictly spreaking there are no data shown in this figure, but mentioned in the caption.

*Action: this figure and its caption have been modified.*

***Minor corrections:***

L. 82 Furthermore -> Furthermore we

*Action: done.*

L. 94 missing ")"

*Action: done.*

L. 104 "(" too much

*Action: done.*

L. 169 Isn't M25 125 Ma age?

*No Action: M25 dates to ~155 Ma in the timescale of Gradstein et al., 2012, which we have used throughout.*

L. 177 "(" too much

*Action: done.*

L. 219-228 This is more "objectivs" than "Data and methods"

*Action: section has been refined.*

L. 229 allows -> allow

*Action: done.*

L. 239 Would be C2 nice to have more detail, seize of the airgun array, length of the streamer...

*Action: More detail has been added (Lines 254-265).*

L. 390 suggested -> suggest Gurnis, M., M. Turner, S. Zahirovic, L. DiCaprio, S. Spasojevic, R. D. Müller, J. Boyden, M. Seton, V. C. Manea, and D. J. Bower, Plate tectonic reconstructions with continuously closing plates, Computers & Geosciences, 38, 35-42, 2012. Hellinger, S. J. (1981). The uncertainties of finite rotations in plate tectonics. Journal of Geophysical Research: Solid Earth, 86(B10), 9312-9318. Chang, T., Ko, D., Royer, J. Y., & Lu, J. (2000). Regression techniques in plate tectonics. Statistical Science, 342-356.

*No Action: These references describe specific tools (Gurnis et al for GPlates, and Hellinger and Chang for one approach to statistical modelling of plate motions from seafloor spreading data) that we have not used at any point for this manuscript and have zero relevance to the discussion of Anomaly J at line 390.*

---

## Author Comment (AC3) · 18 Dec 2019

**Uncertainties in breakup markers along the Iberia-Newfoundland margins illustrated by new seismic data**

Annabel Causer[1], Lucía Pérez-Díaz[1,2], Jürgen Adam[1] and Graeme Eagles[3]

[1]Earth Sciences Department, Royal Holloway University of London, Egham, TW20 0EX, United Kingdom
[2]Department of Earth Sciences, Oxford University, Oxford, OX1 3AN, United Kingdom
[3]Alfred Wegener Institut, Helmholtz Zentrum für Polar und Meeresforschung, Bremerhaven, Germany

Correspondence to: Annabel Causer (annabel.causer.2017@live.rhul.ac.uk)

**Response to reviews**

**Anonymous Referee #3**

The aim of the manuscript submitted by Causer et al. is to discuss breakup markers along the Iberia-Newfoundland margins based on new seismic data. The theme of the manuscript is of major scientific interest, since neither the nature, not the timing and location of breakup are well constrained along the Iberia-Newfoundland margins. Many papers, some of which are very recent, have been dedicated to this problem. I have to admit that I did not find new ideas, or new, well constrained observations that add something new to the subject. Indeed, the interpretation of the new seismic data lack a rigorous interpretation and observations and interpretations are mixed and difficult to follow (for some further comments see comments below). The manuscript reads more like a report referring to old studies and only very few new observations are added. Most disturbing is that some of the latest studies, that come to almost the same conclusions, are only marginally referred or partly not discussed. This omission weighs heavily and discredit the authors. Apart from these points, there are several other points (see comments below) that makes that this manuscript can not be accepted in its present version.

**Specific comments:**

l.30: here and elsewhere in the paper the authors make statements that are similar to the papers of Nirrengarten et al, without citing their work. Actually, most of the conclusions reached in this paper are similar to those by Nirrengarten et al. 2017 and 2018, thus, referring to these results is necessary. I would propose that the authors should discuss how their results are different from those of Nirrengarten et al. 2017 and 2018. I do not really see a big difference. Moreover, the papers of Stanton et al. 2015 and Nirrengarten et al. 2018 that deal with the same subject are not referred to.

*Action: We disagree with the reviewer's statement that our work lacks referencing to that of Nirrengarten on the J-anomaly, although acknowledge our oversight of Nirrengarten et al. 2018, and have now included it.*

*Regarding similarity – both our work and that of Nirrengarten focus on a similar study area and discuss the J-anomaly and its significance for kinematic modelling. However, our work differs from previous studies in that 1) presents and discusses previously unpublished seismic*

*data and 2) illustrates and discusses in detail the impact of "breakup markers" as the basis for plate kinematic modelling.*

l. 25: the SDR packages need to be better described; what is the origin (magmatic) and the significance of the SDR package? I can not find them in the figures

*Action: SDR packages are labelled in figure 5 and described in the text (Section 4).*

l.130 to 150: the tectonic setting part is completely outdated. A lot of work has been done in the last years that need to be referred to.

*No Action: Were this comment to have been more detailed, it would have been difficult to weigh against reviewer 1's suggestion to reduce the level of detail in lines 130-150. As it stands, however, we cannot act on this comment because it lacks any citations to work completed over 'the last few years' that the reviewer thinks we might have missed. With the help of the more detailed and helpful comments made by reviewers 1 and 2, we are confident that this section of the manuscript is both up to date and fit for its purpose.*

l.218 Dataset and methods section need to be rewritten and more details about the data presented in the paper need to be presented.

*No Action: this comment is also too vague as a basis for us to improve our manuscript. The reviewer should have supported their statements with examples and concrete suggestions or advice.*

l.265 to 345 The description of the seismic lines mixes observations with interpretations. Many questions remain open, such as how syn-rift 1 and 2 have been defined, where are possible limits, how were different types of basement defined and what are the evidence for magmatic additions (there are many more questions that arise by looking the seismic interpretations).

*No Action: As no examples are given by the reviewer (and neither of the other two reviewers have highlighted this issue) we are unsure as to where, in the text, the reviewer finds we are mixing observations and interpretations.*

*The rationale for our identification of syn-rift packages, basement and magmatic additions is given in sections 3 and 4.*

The presentation of the data needs to include the presentation of the seismic section (without interpretation), a line drawing and the presentation. As presented here, I cannot follow the interpretations and many of the assumption are not back up by observations. The presentation of the seismic data is insufficient and does not corresponds to the standard of scientific papers.

*No Action: This comment is simply baffling. What figures was the reviewer looking at? Our figures 5 to 7 do in fact include, on separate panels, the seismic section (without interpretation), a line drawing and detail panels, exactly as the reviewer complains they don't.*

l.360 to 400: this section does not really discuss new ideas and does not built on the observations neither. Most of what is said here is old and outdated (the authors seem to have missed the research on the Iberia-Newfoundland margins of the last 5 years??)

*No Action: This section puts our findings (previously presented in section 4) in the context of previously published research. This is the definition of a scientific manuscript's discussion. Regarding referencing, we are unsure as to what research the reviewer is referring to as missing, or out-dated, as yet again no examples are given.*

*We have made changes to this section in response to concrete comments from reviewers 1 and 2.*

l.405 to 460: This section reads more as a report than a discussion chapter.

*No Action: Yet again, a comment that is too vague on its own and too weakly supported by any of the other reviewers' comments to form any no basis on which we can make justifiable changes.*

---

## Author Response (AR2)

Dear Editor,

We thank you for your timely response regarding the revised submission.

Following the advice of Reviewer #1 we have made the following changes to the figures in order to improve the manuscript. The changes we have made are as follows:

*Figure 1 – This figure needs a colour bar, as was the case with figure 1 in the previous version. In addition the 'white envelopes' are nearly impossible to see against the colour scale chosen (GMT ocean?).*

A colour bar has been added to the figure. In addition the 'white envelopes' have been changed to red, and reference to this has been amended in the figure caption.

*Figure 2 – Although it is stated in the responses that the text size has been increased this does not appear to be the case, or if it has it's still too small in my opinion.*

The size of text in all six panels has been increased.

*Figure 3 – As with the previous figure the text size has only slightly been increased and I think it would be better if it was made much larger.*

The size of text has been increased in the figure. Basin abbreviations and line names are now shown in bold.

*Figure 4 – I can confirm that the suggested changes have been made.*

Although Reviewer #1 was happy with the changes to the figure, for clarity we have adapted it slightly. Adaptions include:

(1) Moving the timescale reference from the figure to the figure reference.

(2) Adding further references regarding the poles of rotation used to create the model in the figure reference.

(3) Including abbreviations of the basins along the Newfoundland margin in the figure reference.

[revised manuscript text omitted]